# SuperFi-Cas9 exhibits remarkable fidelity but severely reduced activity yet works effectively with ABE8e

Péter István Kulcsár [1,6], András Tálas[1,6], Zoltán Ligeti[1,2,3], Sarah Laura Krausz[1,4,5] & Ervin Welker [1,2] ✉

Several advancements have been made to SpCas9, the most widely used CRISPR/Cas genome editing tool, to reduce its unwanted off-target effects. The most promising approach is the development of increased-fidelity nuclease (IFN) variants of SpCas9, however, their fidelity has increased at the cost of reduced activity. SuperFi-Cas9 has been developed recently, and it has been described as a next-generation high-fidelity SpCas9 variant, free from the drawbacks of first-generation IFNs. In this study, we characterize the on-target activity and the off-target propensity of SuperFi-Cas9 in mammalian cells, comparing it to first-generation IFNs. SuperFi-Cas9 demonstrates strongly reduced activity but high fidelity features that are in many aspects similar to those of some first-generation variants, such as evo- and HeFSpCas9. SuperFi-cytosine (CBE3) and -adenine (ABE7.10) base editors, as well as SuperFi-prime editor show no meaningful activity. When combined with ABE8e, SuperFi-Cas9, similarly to HeFSpCas9, executes DNA editing with high activity as well as high specificity reducing both bystander and SpCas9-dependent off-target base editing.

Several approaches have been developed[1] to reduce the off-target effects of the SpCas9 nuclease of the type II CRISPR system, which is the most commonly used tool for gene editing applications. Besides approaches that increase the length of the recognition sequence of the nuclease[2–5] or ones that limit its activity in space and/or time[6–9], mutations introduced to the sequence of the ribonucleoprotein complex, affecting either the RNA or the protein component, also seem to be an effective way to increase the specificity of editing. Alteration of the spacer, such as truncation[10] or extension by one[11] or two[12] 5′ G nucleotides as well as the incorporation of modified bases into its sequence, increases the fidelity of SpCas9[13]. Mutations weakening the interactions between the protein and either the targeted[14] or non-targeted DNA strand[15], or the spacer[11], as well as ones weakening the intermolecular interactions between the domains of the protein[16] lead to higher fidelity. Certain amino acid mutations of SpCas9,

identified in selection schemes[17–19], also result in improved discrimination between on-target and off-target sequences. It has been shown that variants with the above-mentioned types of protein mutations become more selective of targets they are active on in exchange for higher fidelity; the higher the fidelity, the more selective the variant is with targets it is active on[11,20–22]. Interestingly, this target-selectivity translates into either fully or partially reduced activity at some targets, while at other targets these increased-fidelity nucleases (IFNs) show wild-type-(WT)-like activity[11,14,16,17,20,21]. Ultimately, their overall average on-target activity is reduced.

A recent paper has provided a very intriguing and comprehensive picture of SpCas9 activation and mismatch tolerance, and has introduced a newly developed variant with the expectation that it can go beyond the above paradigm[23]. The variant was generated by rational design exploiting a cryo-electron microscopy structure of SpCas9 with

[1]Institute of Enzymology, Research Centre for Natural Sciences, Budapest, Hungary. [2]Institute of Biochemistry, Biological Research Centre, Szeged, Hungary. [3]Doctoral School of Multidisciplinary Medical Science, University of Szeged, Szeged, Hungary. [4]Biospiral-2006 Ltd, Szeged, Hungary. [5]School of Ph.D. Studies, Semmelweis University, Budapest, Hungary. [6]These authors contributed equally: Péter István Kulcsár, András Tálas. ✉e-mail: welker.ervin@ttk.hu

the target and single guide RNAs (sgRNAs) mismatching at three consecutive PAM-distal positions: 18–20. The structure revealed that a flexible loop of the RuvC domain stabilizes the distorted end of the target DNA-sgRNA hybrid helix allowing SpCas9 activation even with mismatches. The authors speculated that the residues, stabilizing the distorted helix end, do not participate in interactions in any known SpCas9 structure complexed with on-target DNA. They proposed that by disrupting these mismatch-stabilizing interactions, off-target cleavage activity of SpCas9 can be diminished without affecting on-target cleavage. Based on this, a new increased-fidelity SpCas9 variant was developed by mutating the seven contacting residues to aspartic acids. It was found to exhibit WT-like on-target activity and decreased off-target activity in vitro using a target/off-target pair for which the cryo-electron microscopic structures were made and on which the activity of two IFNs, SpCas9-HF1 and Hypa-SpCas9, had previously been reported to be two orders of magnitude lower than the WT. The authors named this next-generation variant SuperFi-Cas9 (SuperFi for short), inspired by its high fidelity and high on-target activity. They suggested that based on this rationale, further next-generation IFNs may be generated with features distinct to those of the first-generation IFNs that exist to date.

As reported earlier, every IFN exhibits a variety of on-target activity rates in a target-dependent manner, ranging from WT-like on-target activity to close to zero activity[11,14,16,17,20,21]. Whether SuperFi would demonstrate uncompromised on-target activity in general, showing WT-like activity on other targets too is yet unknown. Furthermore, since some of the most important applications of SpCas9 variants exploit its activity in mammalian cells, the most important question raised by the study of Bravo et al. is whether SuperFi will display the proposed features in mammalian cells. We also speculate that the rationale by which SuperFi was generated implies that its mutations will cause reduced activity on off-target sequences where the mismatches are located at positions 18–20, but on sequences with mismatches existing only at other PAM-distal positions, SuperFi will likely show WT-like activity. To answer these questions, we characterize SuperFi for on-target and off-target activity in mammalian cells, comparing it to appropriate first-generation IFNs (Supplementary Fig. 1a). These experiments show that SuperFi possesses high fidelity and a strongly reduced activity level that is typical of the highest fidelity first-generation IFNs, such as evo- and HeFSpCas9. SuperFi is routinely applicable with 21G-sgRNAs, demonstrates high base editing activity with ABE8e and is capable of mitigating its bystander and off-target activities.

## Results

### SuperFi shows strongly reduced activity in cells and in vitro but exhibits high fidelity

The on-target activity of SuperFi was examined in an EGFP disruption assay on 24 genomic targets (Fig. 1a and Supplementary Fig. 1b) and by amplicon sequencing on 26 genomic targets (Fig. 1b and Supplementary Fig. 1c). SuperFi showed significantly lower average on-target activity than WT, Hypa- or SpCas9-HF1, although, on very few targets its activity approached that of the WT nuclease. Hypa- and SpCas9-HF1 reached about 88 and 83%, respectively, while SuperFi reached 15% of the WT activity in the disruption assay. Interestingly, on endogenous targets SuperFi exhibited lower normalized modifications (in median 4% in Fig. 1b; unless otherwise stated percentages throughout the article refer to median values). The difference likely reflects a more saturated condition in the disruption assay. Inhibiting transcription by the binding of SpCas9 to its targets contributes little to this effect in these EGFP disruption experiments, as demonstrated by the dead (inactive) SpCas9 control. Western blot (Supplementary Fig. 1d) and expression plasmid titration analyses (Supplementary Fig. 1e) showed that this lower activity could not be explained by lower expression levels. To assess its fidelity in a mismatch-screen (Fig. 1c) and amplicon

sequencing (Fig. 1d), we selected challenging targets[11,14,16,17,24] that many nuclease variants have failed to edit without off-target activity. Also, we selected targets that have off-targets that contain mismatches exclusively at positions 18–20 or 14–17 according to earlier GUIDE-seq experiments[11,24]. SuperFi showed superior fidelity on many of these targets in comparison to Hypa- and SpCas9-HF1 (Fig. 1c, d and Supplementary Fig. 2). However, contrary to our expectations we saw increased specificity not only at the 18–20 positions but at the 14–17 positions. The genome-wide off-target effects of SuperFi were examined by GUIDE-seq (Fig. 1e, f). Due to its higher fidelity, instead of Hypa- and SpCas9-HF1, we examined SuperFi in comparison with Blackjack-HypaR- and evoSpCas9 that have more similar fidelity to SuperFi[11]. To have meaningful results for these assays we selected targets on which SuperFi (similar to these higher fidelity IFNs) is expected to exhibit a reasonable on-target activity. This rationale is based on our recent study[25] showing that targets have different cleavability, and low activity/high-fidelity IFNs can cleave only high cleavability targets. SuperFi exhibited higher specificity than Blackjack-HypaR- and evoSpCas9 on 3 and 2 of the 4 examined targets, respectively (Fig. 1e, f and Supplementary Fig. 3).

We wondered if SuperFi's strongly reduced activity could be detected in in vitro experiments as well. Thus, we examined its in vitro activity in a plasmid cleavage assay on 8 targets from Fig. 1a, on which SuperFi showed WT-like (3 targets) or zero/almost zero (5 targets) activity in cells. These experiments revealed that SuperFi also showed strongly reduced activity in vitro, with its cleavage activity approaching that of WT on those targets on which it also had WT-like activity in cells (Fig. 1g and Supplementary Fig. 4).

### Characterization of SuperFi's activity with 21G-sgRNAs and the effect of the individual mutations of SuperFi

5′ extended or truncated spacers are frequently applied with SpCas9 nucleases, as these can increase the fidelity of editing[5,10–12,26], and a 5′ G extension of the spacer (21G-sgRNA) is often necessary to comply with the sequence preference of the U6 and T7 promoters often used for the expression of sgRNAs[11,18]. Only Sniper Cas9[18], the lowest fidelity IFN[11], can be routinely used with 5′ truncated sgRNAs, and only Sniper and the Blackjack variants can be routinely used with 21G-sgRNAs amongst the first-generation IFNs[11,18]. Although truncating the sgRNAs diminished the activity of SuperFi (Supplementary Fig. 5a), it seems to be fully compatible with the extended 21G-sgRNAs (Fig. 2a, b and Supplementary Fig. 5b). We confirmed these results by comparison with evoSpCas9, using both 20G- and 21G-sgRNAs on six endogenous targets on which evoSpCas9 was known to have reasonable activity with 20G-sgRNAs. However, the targets were only cut by SuperFi with 21G-sgRNAs, albeit with lower efficiency on some targets than we saw in the EGFP disruption experiments (Fig. 2c, d).

Furthermore, we aimed to elaborate the specific contribution of SuperFi mutations to its increased-fidelity/decreased activity, thus we engineered seven new variants, each lacking a single mutation of SuperFi. Figure 2e (and Supplementary Fig. 6) shows that the mutations have largely equal contributions, except for the 1010D mutation, which actually seems to increase activity [thus the SuperFi (D1010Y) variant shows decreased activity compared to SuperFi], and the 1031D mutation, which has a larger activity-decreasing contribution than the others. In a mismatch-screen we examined this SuperFi (D1031K) variant, which showed that the absence of the 1031D mutation results in a loss of fidelity along with an increase in activity compared to SuperFi (Fig. 2f and for source data see Supplementary Data 2).

### SuperFi is not compatible with cytosine base editor 3 and adenine base editor 7.10

Based on the high fidelity of SuperFi, we proposed that it may be particularly effective in decreasing the off-target effects of base[27–30]

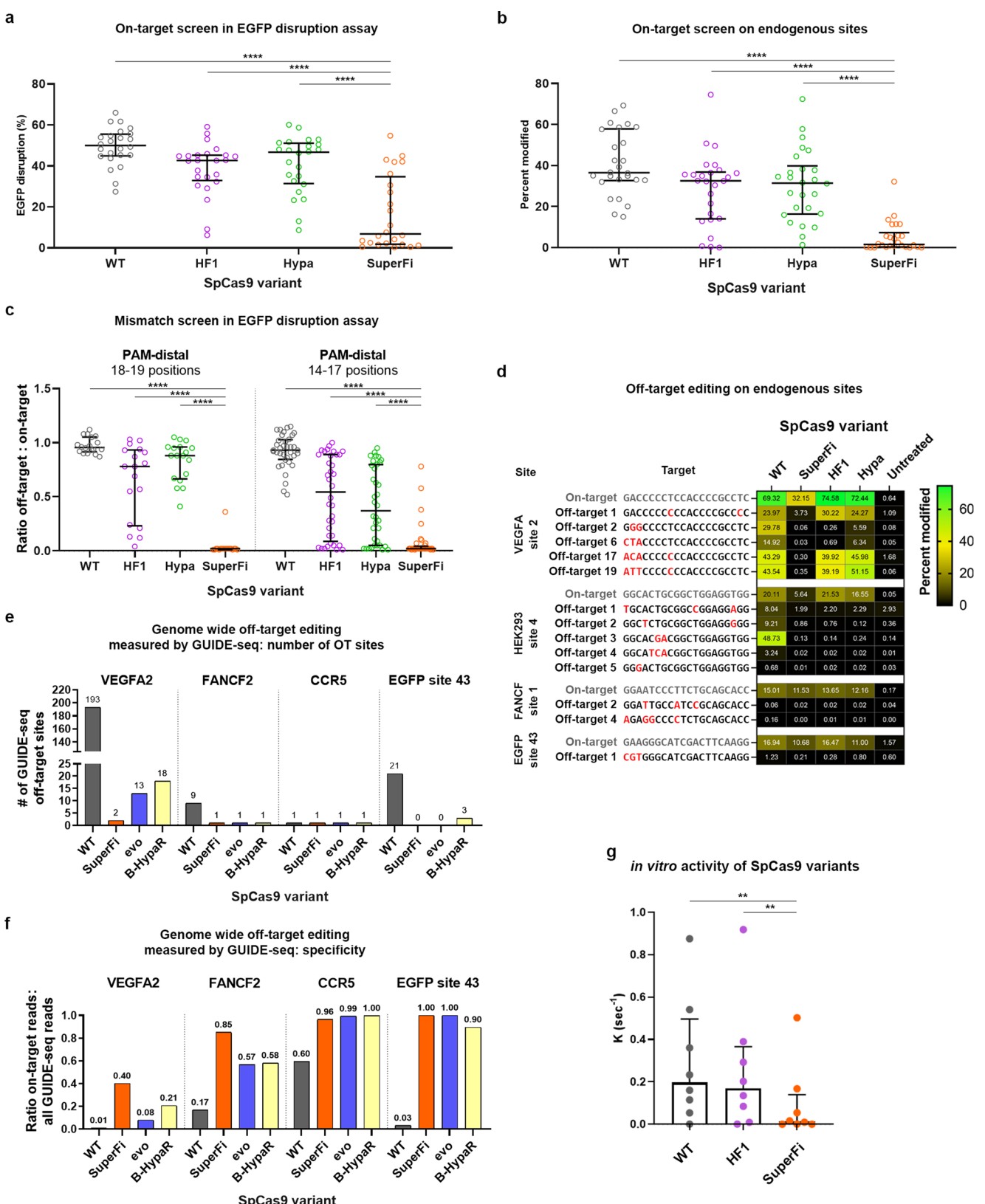

and prime editing[31]. For these experiments we utilized two recently developed plasmid-based fluorescent assays, BEAR[32] and PEAR[33] which are suitable for assessing base and prime editing activity, respectively. Both assays are based on a reporter GFP sequence interrupted by an intron with an inactive splice donor site. When the splice site is corrected by the specific action of base editing or prime editing (indels do

not correct the splice site) transiently fluorescent cells can be detected. We examined SuperFi in these experiments along with those IFNs showing the closest activity/fidelity in a previous study[32] aimed at elaborating the mechanism of IFN base editors. When tested, SuperFi cytosine base editor 3 (CBE for short) and SuperFi adenine base editor 7.10 (ABE7 for short) exhibited strongly reduced base editing activity

**Fig. 1 | On- and off-target activities of the SuperFi-Cas9 nuclease in mammalian cells and in vitro. a**, **b** On-target activity of various SpCas9 variants as measured **a** in EGFP disruption assay on 24 or **b** by NGS on 26 endogenous target sites presented on a scatter dot plot. Data are also presented in Supplementary Fig. 1b and c, respectively. **c** Off-target activity of various SpCas9 variants as measured in EGFP disruption assay presented on a scatter dot plot. Only those data points are presented here for which the on-target activity exceeded 70%. **d** Off-target activity of various SpCas9 variants as measured by NGS on endogenous target sites presented on a heatmap. The heatmap shows the mean on- and off-target modifications (indels) of three parallel transfections. Data related to endogenous on-target sites where editing was low (<5%) with SuperFi-Cas9 are not shown on this figure, but data are available in Supplementary Data 3. On- and off-target editing was measured from the same samples. The mismatching nucleotides at DNA off-target sites are indicated as red letters. **c**, **d** Data are also presented in Supplementary Fig. 2a and b, respectively. **e**, **f** Bar charts of **e** the total number of off-target sites and **f** the on-target cleavage specificity, expressed as the percentages of the on-target reads from all reads, as measured by GUIDE-seq. Data are also presented in Supplementary Fig. 3. **g** In vitro cleavage activities of the variants employing 8 targets of panel **a**. Data are also presented in Supplementary Fig. 4. The median and interquartile range are shown; data points represent the mean of the fitted k value triplicates. **a**–**c** The median and interquartile ranges are shown; data points are plotted as open circles representing the mean of triplicates. **a**–**g** Summary of target and primer sequences, EGFP disruption, NGS, GUIDE-seq and in vitro data are reported in Supplementary Data 1–5. Statistical significance was assessed by RM one-way ANOVA, statistical details and $p$-values are available in Methods and in Supplementary Data 6 (*$p < 0.05$, **$p < 0.01$, ***$p < 0.001$, ****$p < 0.0001$).

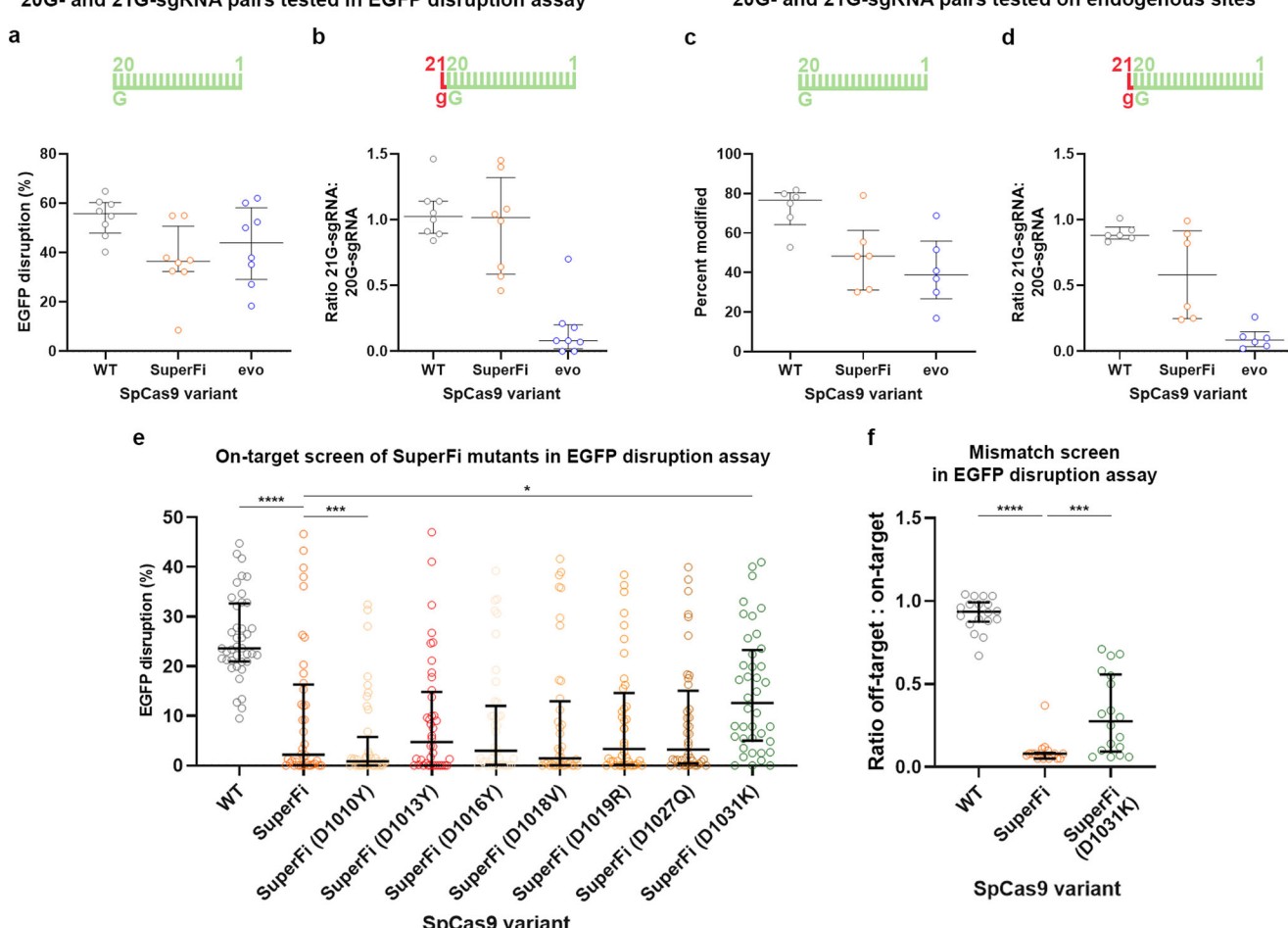

**Fig. 2 | Characterization of SuperFi and its variants in mammalian cells. a**–**d** (**a**, **b**) EGFP disruption activity on 8 target sites and **c**, **d** indels measured on 6 endogenous sites with **a**, **c** 20G-sgRNAs and the same sites targeted with **b**, **d** 21G-sgRNAs, normalized to their 20G-sgRNA counterparts, presented on a scatter dot plot. Data are related to Supplementary Fig. 5b, c. **e** On-target activity of various SpCas9 variants as measured in EGFP disruption assay on 40 target sites presented on a scatter dot plot. Data are related to Supplementary Fig. 6. **f** Normalized EGFP disruption activity of SpCas9 nucleases with perfectly matching ($n = 18$) and partially mismatching ($n = 162$) 20G-sgRNAs. Dots are shown for each variant with each mismatching spacer position, provided that the on-target activity exceeded 70% with every SpCas9 variant; data are omitted otherwise. **a**–**f** The median and interquartile ranges are shown; data points are plotted as open circles representing the mean of triplicates. Summary of target and primer sequences, EGFP disruption and NGS data are reported in Supplementary Data 1–3. Statistical significance was assessed by RM one-way ANOVA, statistical details (only statistically significant differences compared to SuperFi are shown) and $p$-values are available in Methods and in Supplementary Data 6 (*$p < 0.05$, **$p < 0.01$, ***$p < 0.001$, ****$p < 0.0001$).

both in the BEAR assay (17 and 15%) and on endogenous targets (29 and 9%) normalized to WT CBE and WT ABE7, respectively (Fig. 3a, Supplementary Figs. 7 and 8). However, SuperFi-ABE8e exhibited more substantial activity both in the BEAR fluorescent assay (Fig. 3a and Supplementary Fig. 7a), and on genomic targets (Fig. 3b and Supplementary Fig. 7b), with 84 and 56% median editing normalized to the WT ABE8e, respectively.

In a former study[32] we examined IFN-CBEs and IFN-ABEs on the same target sets using the BEAR assay. This experimental design facilitates the comparison of CBE and ABE base editing activities under

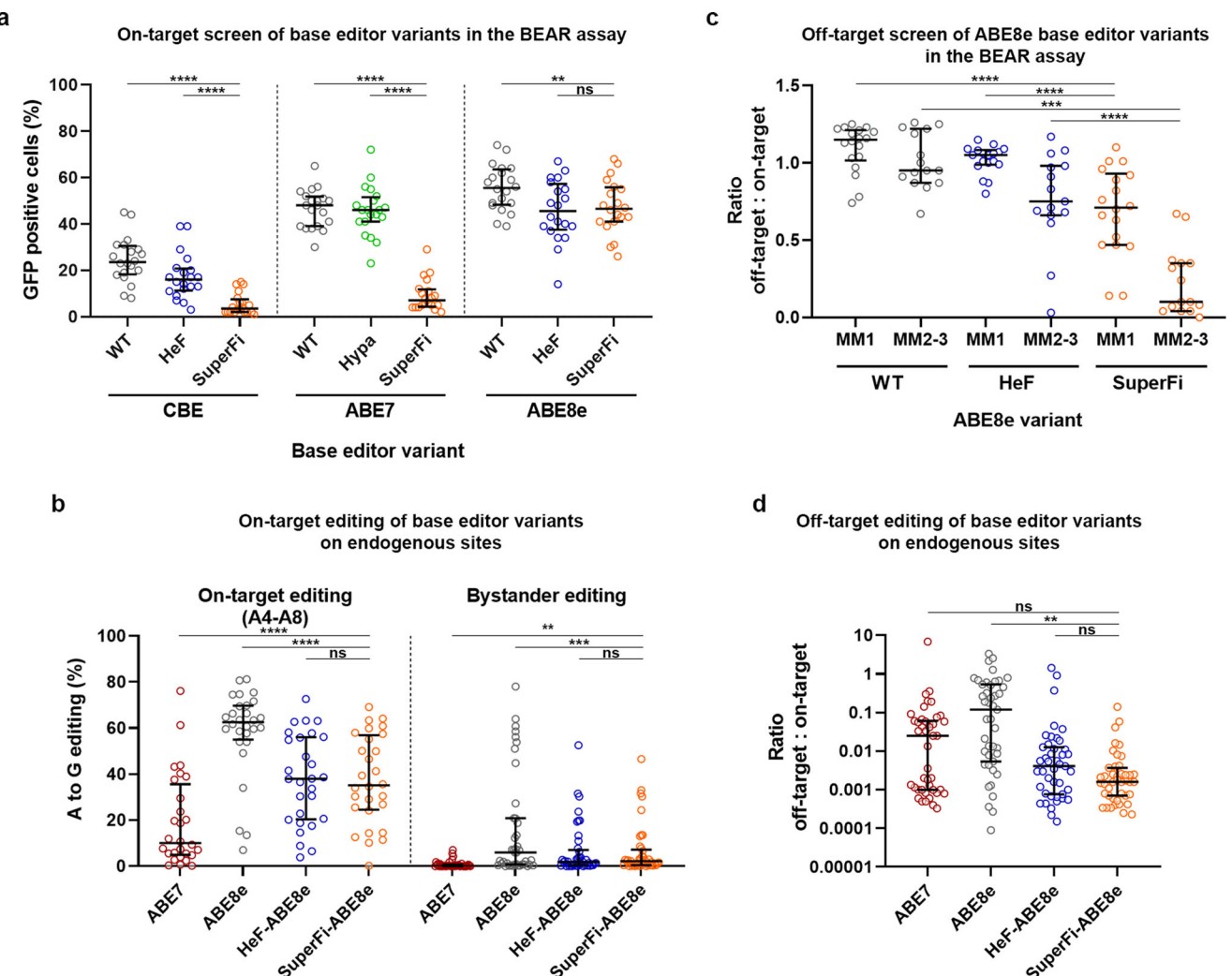

**Fig. 3 | On- and off-target activities of SuperFi base editors in mammalian cells.**
**a**, **c** Base editing activities of CBE, ABE7, and ABE8e variants were assessed in the BEAR assay with **a** 20 matching and **c** 33 mismatching sgRNAs. The same data are shown by individual targets in Supplementary Figs. 7a and 9a, respectively.
**b**, **d** Base editing activities of ABE8e variants on endogenous **b** on-target and **d** off-target sites as measured by NGS. Data are also presented in Supplementary Figs. 7b and 9b, respectively. **b** The editing efficiency (16 genomic sites) of adenines inside (A4-A8, $n = 28$) and outside the editing window (bystander editing, $n = 41$) is shown side by side, separated with a dashed line. Regarding base editing, the numbering of the edit positions follows the convention laid in the literature, i.e., 5′ to 3′ direction in the non-targeted strand of the target DNA. **c** The fidelity of ABE8e variants was assessed in the BEAR assay with 2 matching sgRNAs and 33 sgRNAs mismatching in one (MM1; $n = 18$) or in two to three positions (MM2-3; $n = 15$). The relative activity of all base editors (off-target/on-target ratio) for all adenines is plotted separately for all MM1 and all MM2-3 sgRNAs. **d** The relative activity (off-target/on-target ratio) of ABE variants is shown for all adenines ($n = 43$) of 17 genomic off-target sites. For on-target adenine value, we selected the adenine which was edited by ABE7 (for EMX1 by ABE8e) at the highest level for each on-target sequence. **a**–**d** Results are presented on a scatter dot plot, the median and interquartile ranges are shown; data points are plotted as open circles representing the means of triplicates. Summary of target and primer sequences, BEAR, NGS data and allele tables are reported in Supplementary Data 1–3 and 7. Statistical significance was assessed by RM one-way ANOVA, statistical details and $p$-values are available in Methods and in Supplementary Data 6 (*$p < 0.05$, **$p < 0.01$, ***$p < 0.001$, ****$p < 0.0001$).

conditions when SpCas9 encounters the same interactions with the targets, i.e.,: the same extent/transiency of R-loop formation. These experiments revealed that higher fidelity IFNs such as evo- and HeFSpCas9 were not active in combination with ABE7 on-targets on which they were active in combination with CBE, while they turned to be active with ABE8e which has a deaminase variant with orders of magnitude higher activity. Altogether, these findings suggested that the relation between the rate of deamination and the extent/transiency of R-loop formation determines how an IFN alters on- and off-target activity of base editors (the factors affecting base editors' activity is summarized in ref. 34). Along this line of thought, we tested whether using a faster rAPOBEC deaminase variant, such as the one present in evoCBE[34] would increase the base editing activity of SuperFi. Although increased, SuperFi-evoCBE activity still remained low in both BEAR (2%) and on endogenous targets (12.7%) (Supplementary Fig. 8a, b).

The decreased ability of SuperFi to function as a CBE variant in comparison to evo- and HeFSpCas9 calls for an investigation on the extent/transiency of its R-loop formation that may be different from those of evo- and HeFSpCas9. Since neither SuperFi-ABE7 (1%, see Supplementary Fig. 7c), nor SuperFi-CBE (8%, see Supplementary Fig. 8b) seem to have meaningful activities, we did not attempt to assess their off-target propensity.

**Effective and specific adenine base editing with SuperFi-ABE8e**
Due to its high, undiscriminating activity, ABE8e had been used in only a few applications. However, SuperFi-ABE8e was found to be active and could successfully reduce SpCas9-dependent off-target effects when assessed either in the BEAR assay, (2.4- and 9.5-fold more reduced on the 1 and the 2–3 mismatches, respectively) (Fig. 3c and Supplementary Fig. 9a), or on genomic targets (74-fold more specific) (Fig. 3d and

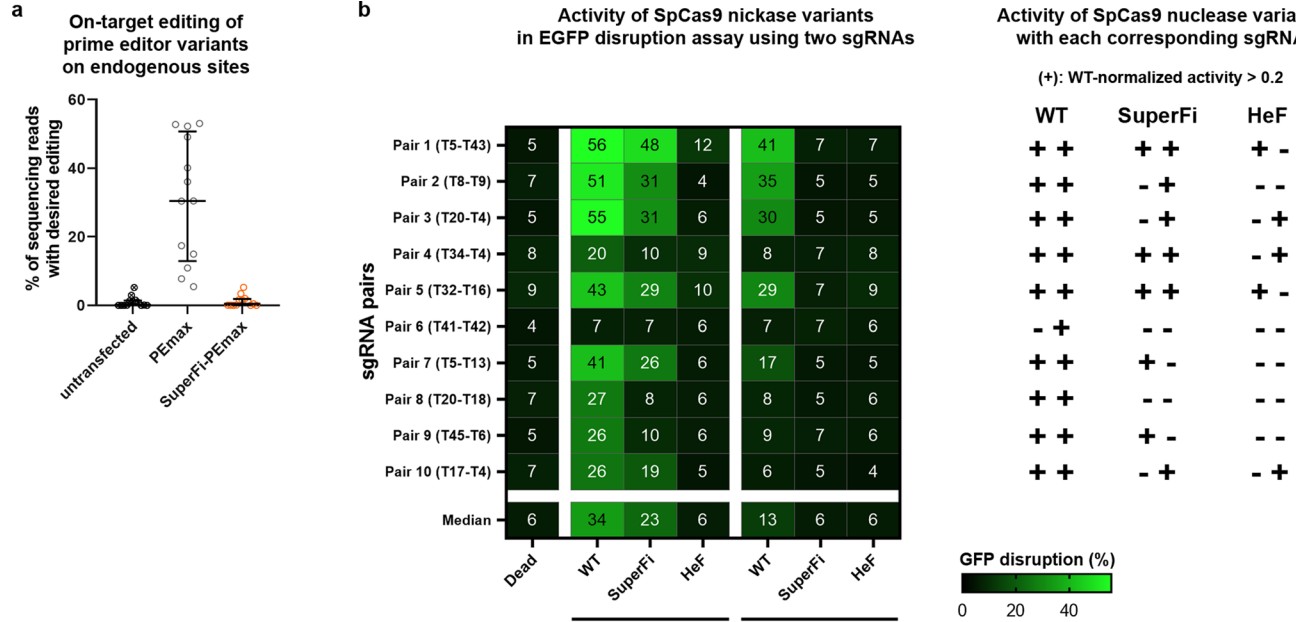

**Fig. 4 | On-target activities of SuperFi prime editor and nickase activities of D10A and H840A nickase variants in mammalian cells. a** Prime editing activity of SuperFi. Data are from Supplementary Fig. 10d–e. Results are presented on a scatter dot plot, the median and interquartile ranges are shown; data points are plotted as open circles representing the means of triplicates for n = 13 target sites. **b** (left) On-target disruption activities of nickase SpCas9 variants with paired sgRNAs. The heatmap shows the mean on-target modifications (indels) of three parallel transfections. (right) +/− indicate the activity of the nuclease variants with the corresponding sgRNAs. **a, b** Summary of target and primer sequences, EGFP disruption and NGS data are reported in Supplementary Data 1–3.

Supplementary Fig. 9b). It also demonstrated a lower level of bystander editing compared to WT-SpCas9 fused with ABE8e (2.7-fold more specific), while maintaining significantly higher on-target activity compared to ABE7 (2.4-fold more active and 9.5-fold more specific) (Fig. 3b, Supplementary Fig. 7b) on genomic targets. In addition, we compared SuperFi-ABE8e with WT-ABE7 and WT-ABE8e to test if its activity window is shifted. The highly similar editing pattern on the same targets suggests that the editing window is not shifted, and only the activity is reduced compared to ABE8e (Supplementary Fig. 10a, b, for allele frequency tables see Supplementary Data 7). When SuperFi- and HeF-ABE8e were compared, no significant differences were detected in their on-target, off-target and bystander activities.

## SuperFi is not active as a prime editor

SuperFi in combination with a prime editor (PEmax 3[35]) was unsuccessful, as it showed little to no activity in the PEAR[33] fluorescent assay (Supplementary Fig. 10c) and showed no activity at all on 10 genomic loci installing 13 types of edits (Fig. 4a and Supplementary Fig. 10d, e). To assess the reason underlying its inactivity, in a double nickase disruption assay we tested if this issue is related to the combined effect of the seven SuperFi and the R221K, N394K mutations present in PEmax. These lysine mutations are supposed to increase the nuclease activity[36], and in PEmax these mutations may increase the nickase activity of SpCas9. Interestingly, the presence or absence of the R221K, N394K mutations did not affect the nickase activity of WT prime editors here. Moreover, neither SuperFi-PE, nor SuperFi-PEmax showed nickase activity on any of the target pairs (Supplementary Fig. 10f). To see whether the reverse transcriptase partner interfered with SuperFi's nickase activity, we examined the activity of both the D10A and the H840A SuperFi nickases along with HeF and WT SpCas9s. Interestingly, for WT SpCas9, both nickases showed high activity, while for SuperFi-Cas9 only the D10A nickase showed detectable activity. As it is known that the D10A nickase SpCas9 variant has higher activity than the H840A nickase has[37], it is not surprising that the D10A SuperFi variant was found to be more active here as well (Fig. 4b). However, the

non-detectable activity of SuperFi-H840A may (at least partly) provide an explanation for the very low prime editing activity of SuperFi.

## Discussion

Several factors could explain why SuperFi did not show the features suggested by the study of Bravo et al.[23]. One may propose that the mutations within the RuvC loop interfere with cellular factors that specifically diminish its high in vitro activity in mammalian cells. Alternatively, SuperFi may not be expressed at the same level as other IFNs. Neither the fact that SuperFi has WT-like activity on a few targets, nor the western blot and plasmid dilution experiments support this scenario. Indeed, SuperFi also shows low on-target activity in vitro, even lower than SpCas9-HF1 on this set of targets. Thus, a likely explanation is that the single target used in ref. 23 may happen to be one of the few targets on which SuperFi can demonstrate an activity rate higher than SpCas9-HF1, close to that of WT-SpCas9.

Experiments with each variant lacking a single mutation (Fig. 2e, f) revealed that by selecting different combinations of the mutations of SuperFi, variants with higher on-target activity but proportionally lower fidelity can be generated. In this respect, SuperFi is very similar to first-generation IFNs, especially to those with the highest fidelity, such as evo-, B-HypaR- and HeFSpCas9[11,20,32]. Somehow, the rationale underlying its development does not seem to be manifested in the features of SuperFi: the RuvC loop mutations in the engineered variant do not specifically act on the targets with 18–20 mismatches, leaving on-targets and targets with mismatches at other positions unaffected.

Instead, these experiments suggest a possible role for the RuvC loop in both the on- and off-target activities of SpCas9. SuperFi shares features of Blackjack IFNs, which are all active with 21G-sgRNAs. This may relate to the fact, that RuvC loop residues such as 1013Tyr (first described in ref. 38), which is aligned to the 5' end of sgRNA in one of the two molecules in the asymmetric unit of PDB4OO8[38], appear to cap the sgRNA. These capping interactions may provide a structural explanation for the activity-reducing, fidelity-increasing effect of 5'-extension of the sgRNAs[11,12], that is more pronounced in (non-

Blackjack) IFNs than in WT. Alterations in the RuvC loop may disrupt these interactions, and thus reduce the effect of the 5′ extension of the sgRNA on the activity of SpCas9.

In ref. 11, we reported on constructing 19 variants with mutations altering the length and dynamics of the RuvC loop. We aimed to find a modification of this loop that minimally reduces on-target activity but makes IFNs tolerate 21G-sgRNAs. All these variants decreased the on-target activity of SpCas9, pointing to the important role this loop may have in the functioning of SpCas9. These former experiments showed that alterations of the RuvC loop may decrease on-target activity but increase fidelity, resulting in novel IFNs. Our data presented here on the on-target activity and specificity, as well as on the activity with 21G-sgRNAs of SuperFi further support this scenario. We expect that further structural and mechanistic research will reveal the exact role of this RuvC loop in SpCas9 function.

Although its relatively low on-target activity does not make SuperFi useful for general use with various applications, in the case of the super-active ABE8e it seems to effectively counteract the over-activity of the mutant deaminase partner, similarly to other lower activity/higher fidelity SpCas9 variants, such as HeFSpCas9. Thus, they behave as a very effective tool that are significantly more active than ABE7 and more specific than ABE8e.

## Methods
### Materials
Restriction enzymes, T4 ligase, Dulbecco's modified Eagle Medium DMEM (Gibco), fetal bovine serum (Gibco), Turbofect, Qubit dsDNA HS Assay Kit, Taq DNA polymerase (recombinant), TranscriptAid T7 High Yield Transcription Kit, Platinum Taq DNA polymerase, 0.45 μm sterile filters and penicillin/streptomycin were purchased from Thermo Fischer Scientific. DNA oligonucleotides, trimethoprim (TMP) and GenElute HP Plasmid Miniprep kit were acquired from Sigma–Aldrich. ZymoPure Plasmid Midiprep, RNA Clean & Concentrator kit and Maxiprep kits were purchased from Zymo Research. NEBuilder HiFi DNA Assembly Master Mix and Q5 High-Fidelity DNA Polymerase were obtained from New England Biolabs Inc. NucleoSpin Gel and PCR Clean-up kit was purchased from Macherey-Nagel. SF Cell Line 4D-Nucleofector X Kit S were purchased from Lonza, Bioruptor 0.5 ml Microtubes for DNA Shearing from Diagenode. Agencourt AMPure XP beads were purchased from Beckman Coulter. T4 DNA ligase (for GUIDE-seq) and end-repair mix were acquired from Enzymatics. KAPA universal qPCR Master Mix was purchased from KAPA Biosystems.

### Plasmid construction
SuperFi-Cas9 vectors were constructed using NEBuilder HiFi DNA Assembly. All SpCas9, base and prime editor variants were subcloned to their corresponding expression plasmid backbones. For detailed cloning and sequence information see Supplementary Information. sgRNA coding plasmids were constructed as detailed in Supplementary Information. The list of sgRNA target sites, mismatching sgRNA sequences and plasmid constructs used in this study are available in Supplementary Data 1. The sequences of all plasmid constructs were confirmed by Sanger sequencing (Microsynth AG).

In all experiments the following plasmids were used for SpCas9, base and prime editor variant expression (Addgene# provided): pX330-Flag-WT_SpCas9 (without sgRNA; with silent mutations) (#126753), pX330-Flag-SpCas9-HF1 (without sgRNA; with silent mutations) (#126755), pX330-Flag-HypaSpCas9 (without sgRNA; with silent mutations) (#126756), pX330-Flag-evoSpCas9 (without sgRNA; with silent mutations) (#126758)[11], B-HypaR-SpCas9 (#126764)[25], pAT9676-ABE (#162997) for ABE7, pAT9749-dABE (#162998) for deadABE7, pAT9993-Hypa-ABE (#163001) for Hypa-ABE7; pAT9675-CBE (#163007) for CBE; pAT15069-HeF-CBE (#163007) for HeF-CBE; pAT15482_ABE8e (#174120) for ABE8e; pAT15488_HeF-ABE8e

(#174126) for HeF-ABE8e[32] and pCMV-PEmax[35] (#174820) for PEmax expression. pET-Cas9-NLS-6xHis (Addgene #62933)[39], pET-SpCas9-HF1-NLS-6xHis, pET-SuperFi-Cas9-NLS-6xHis were used for WT, -HF1 and SuperFi-SpCas9 bacterial expression, respectively. Plasmid constructs coding SuperFi (D1010Y), SuperFi (D1013Y), SuperFi (D1016Y), SuperFi (D1018V), SuperFi (D1019R), SuperFi (D1027Q), SuperFi (D1031K) were used for testing new SuperFi mutants.

Plasmids developed by us in this study and deposited at Addgene are the following: pPIK16045_pX330_Flag-SuperFi-Cas9 (#184370), pAT15542_nCBE-SuperFi-Cas9 (#184372), pAT15544_nABE-SuperFi-Cas9 (#184374), pAT15543_dABE-SuperFi-Cas9 (#184373), pAT15546_nABE8e-SuperFi-Cas9 (#184376), pAT15545_dABE8e-SuperFi-Cas9 (#184375), pAT15547_PEmax-SuperFi-Cas9 (#184377).

### Cell culturing and transfection
Cells employed in this study are HEK293 (Gibco 293-H cells), N2a.dd-EGFP (a neuro-2a mouse neuroblastoma cell line developed by us containing a single integrated copy of an EGFP-DHFR[DD] [EGFP-folA dihydrofolate reductase destabilization domain] fusion protein coding cassette originating from a donor plasmid with 1,000 bp long homology arms to the *Prnp* gene driven by the *Prnp* promoter (*Prnp*.HA-EGFP-DHFR[DD]), N2a.EGFP and HEK-293.EGFP (both cell lines containing a single integrated copy of an EGFP cassette driven by the *Prnp* promoter)[20] cells. Cells were grown at 37 °C in a humidified atmosphere of 5% $CO_2$ in high glucose Dulbecco's Modified Eagle medium (DMEM) supplemented with 10% heat inactivated fetal bovine serum, 4 mM L-glutamine (Gibco), 100 units/ml penicillin and 100 μg/ml streptomycin. Cells were passaged up to 20 times (washed with PBS, detached from the plate with 0.05% Trypsin-EDTA and replated). After 20 passages, cells were discarded. Cell lines were not authenticated as they were obtained directly from a certified repository or cloned from those cell lines. Cells were tested for mycoplasma contamination.

Cells were plated in case of each cell line one day prior to transfection in 48-well plates at a density of $3 \times 10^4$ cells/well. The following amounts of plasmids were mixed with 1 μL Turbofect reagent in 50 μL serum-free DMEM and were incubated for 25–30 minutes prior to being added to the cells: (1) in case of SpCas9 cleavage (either in EGFP disruption or in genomic editing): SpCas9 variant expression plasmid (137 ng) (for titration, the amount was always supplemented with dead NmCas9 expression plasmid to 137 ng) and sgRNA and mCherry coding plasmid (97 ng) (for the double nicking experiments 48.5–48.5 ng of both sgRNA coding plasmids was added); (2) in case of genomic base editing: base editor variant expression plasmid (190 ng) and sgRNA and BFP coding plasmid (83 ng); (3) in case of genomic prime editing: prime editor variant expression plasmid (222 ng), pegRNA and BFP coding plasmid (55 ng), and second nicking sgRNA and BFP coding plasmid (36 ng).

EGFP disruption experiments were conducted in N2a.EGFP cells for the on-target screen. In this cell line the EGFP disruption level is not saturated, this way this assay is a more sensitive reporter of the intrinsic activities of these nucleases compared to N2a.dd-EGFP cell line. EGFP disruption experiments were conducted in N2a.EGFP cells for the mismatch screen, 21G- and truncated sgRNA screen. In this cell line four days post-transfection results show a close to saturated level, this way it is a good reporter system for seeing the full spectrum of activities[11,20].

For Fig. 2f mismatch-screen N2a.dd-EGFP cells were co-transfected with two types of plasmids: an SpCas9 variant expression plasmid (137 ng) and a mix of 3 sgRNAs in which one nucleotide position was mismatched to the target using all 3 possible bases and mCherry coding plasmid ($3 \times \sim 33.3$ ng = 97 ng) using 1 μl TurboFect reagent per well in 48-well plates. TMP (trimethoprim; 1 μM final concentration) was added to the media ~48 h before FACS analysis. Transfected cells were analyzed four days post-transfection by flow cytometry. For this cell line, the 4-day post-transfection results showed

an almost saturated level, thus it is a good reporter system to capture the full spectrum of off-target activities. By using a mixture of the three sgRNAs including all the three possible mismatching bases in the tested position, makes it straightforward to screen more positions and get more balanced off-target propensities for each[20].

The BEAR and PEAR reporters were used as described previously[32,33]. Briefly, cells were transfected as described above where in case of BEAR experiments: 66 ng of BEAR target plasmid, 56 ng of sgRNA-mCherry and 127 ng of base editor coding plasmid were used. In case of PEAR experiments: 40 ng of PEAR-GFP-2in1 target plasmid, 56 ng of mock pegRNA-BFP (to follow transfection efficiency) and 255 ng of prime editor coding plasmid were used.

Cells were analyzed by flow cytometry three days post-transfection (-60–75 h) in case of BEAR and PEAR experiments and four days post-transfection (-80–100 h) in all other experiments. Transfections were performed in triplicates. Transfection efficacy was calculated via mCherry or BFP expression. Data of the EGFP disruption, BEAR and PEAR experiments are available in Supplementary Data 2, Source data.

### Electroporation in GUIDE-seq experiments

Briefly, $2 \times 10^5$ cells were resuspended in transfection solution (see below) and mixed with 666 ng of SpCas9 variant expression plasmid and 334 ng of sgRNA and mCherry coding plasmid and an additional 30 pmol dsODN (according to the original GUIDE-seq protocol[24]) was added to the mixture. Nucleofections were performed in the case of HEK293 and HEK-293.EGFP cell lines using the CM-130 program on a Lonza 4-D Nucleofector instrument on strip with 20 µl SF solution according to the manufacturer's protocol. Transfections were performed in triplicates. Transfection efficacy was calculated via mCherry expression.

### Flow cytometry

Flow cytometry analyses were carried out on an Attune NxT Acoustic Focusing Cytometer (Applied Biosystems). For data analysis Attune NxT Software v.4.2 was used. Viable single cells were gated based on side and forward light-scatter parameters and a total of 5000–10,000 viable single cell events were acquired in all experiments. BFP, GFP, and mCherry signals were detected using the 405 (for BFP), 488 (for GFP) and 561 nm (for mCherry) diode laser for excitation, and the 440/50 (BFP), 530/30 (GFP) and 620/15 (mCherry) filter for emission. For detailed flow cytometry gating information see Supplementary Fig. 11.

### Indel, base, and prime editing analysis by next-generation sequencing (NGS)

Transfected cells were analyzed by flow cytometry (to assess transfection efficiency) followed by genomic DNA purification according to the Puregene DNA Purification protocol (Gentra systems). Amplicons for next-generation sequencing were generated from the genomic DNA samples using two rounds of PCR to attach Illumina handles. The 1st step PCR primers used to amplify target genomic sequences are listed in Supplementary Data 1: NGS primers. PCR was done in a S1000 Thermal Cycler (Bio-Rad) or PCRmax Alpha AC2 Thermal Cycler using Q5 high-fidelity polymerase supplemented with Q5 buffer (in case of VEGFA site 2 amplicon supplemented with Q5 High GC enhancer as well) and 150 ng of genomic DNA in a total volume of 25 µl. The thermal cycling profile of the PCR was: 98 °C 30 sec; 35 × (denaturation: 98 °C 20 sec; annealing: see Supplementary Data 1: NGS primers, 30 sec; elongation: 72 °C, see Supplementary Data 1: NGS primers); 72 °C 5 min. i5 and i7 Illumina adapters were added in a second PCR reaction using Q5 high-fidelity polymerase with supplied Q5 buffer (in case of VEGFA site 2 amplicon together with Q5 High GC enhancer) and 1 µl of first step PCR product in total volume of 25 µl. The thermal cycling profile of the PCR was: 98 °C 30 sec; 35 × (98 °C 20 sec, 67 °C 30 sec, 72 °C 20 sec); 72 °C 5 min. Amplicons were purified by agarose gel electrophoresis. Samples were quantified with Qubit dsDNA HS Assay

kit and pooled. Double-indexed libraries were sequenced on a MiniSeq or NextSeq (Illumina) giving paired-end sequences of 2 ×150 bp, performed by Deltabio Ltd. Reads were aligned to the reference sequence using BBMap.

Indels were counted computationally amongst the aligned reads that matched at least 75% of the first 20 bp of the reference amplicon. Indels without mismatches were searched starting at ±2 bp around the cut site. For each sample, the indel frequency was determined as (number of reads with an indel) / (number of total reads). Frequency of substitutions without indels generated by base or prime editing was determined as the percentage of (sequencing reads with the intended modification, without indels) / (number of total reads). Allele frequency tables were generated using CRISPResso2[40]. By contrast, frequency of intended insertions or deletions generated by prime editing was determined as the percentage of (all sequencing reads with only the intended insertions or deletions) / (number of total reads). For these samples the indel background was calculated from reads containing types of indels that were different from the aimed edit. The 15 bp long center fragment of the GUIDE-seq dsODN sequence ("gttgtcatatgttaa" / "ttaacatatgacaac") was counted in the aligned reads to measure dsODN on-target tag integration for GUIDE-seq experiments.

The following software were used: BBMap 38.08, samtools 1.8, BioPython 1.71, PySam 0.13. For NGS data information see Supplementary Data 3. The deep sequencing data are available in NCBI Sequence Read Archive (accession number: PRJNA876837).

### GUIDE-seq

In the first step genomic DNA was sheared with BioraptorPlus (Diagenode) to 550 bp in average. Sample libraries were assembled as previously described[24] and sequenced on Illumina MiniSeq instrument by Deltabio Ltd. Data were analyzed using open-source GUIDE-seq software (version 1.1)[41]. Consolidated reads were mapped to the human reference genome GrCh37 in case of EGFP target site 43 supplemented with the integrated EGFP sequence. Upon identification of the genomic regions integrating double-stranded oligodeoxynucleotide (dsODNs) in aligned data, off-target sites were retained if at most seven mismatches against the target were present and if absent in the background controls. Visualization of aligned off-target sites are provided as a color-coded sequence grid. Summarized results can be found in Supplementary Data 4, Source data and GUIDE-seq sequencing data are available in NCBI Sequence Read Archive (accession number: PRJNA876837).

### Western blot

N2a.dd-EGFP cells were cultured on 48-well plates and were transfected as described above in the EGFP disruption assay section. Four days post-transfection, 9 parallel samples corresponding to each SpCas9 variant transfected were washed with PBS, then trypsinized and mixed, and were analyzed for transfection efficiency via mCherry fluorescence level by using flow cytometry. The cells from the mixtures were centrifuged at $200 \times g$ for 5 min at 4 °C. Pellets were resuspended in ice cold Harlow buffer (50 mM Hepes pH 7.5; 0.2 mM EDTA; 10 mM NaF; 0.5% NP40; 250 mM NaCl; Protease Inhibitor Cocktail 1:100; Calpain inhibitor 1:100; 1 mM DTT) and lysed for 20–30 min on ice. The cell lysates were centrifuged at $19,000 \times g$ for 10 min. The supernatants were transferred into new tubes and total protein concentrations were measured by the Bradford protein assay. Before SDS gel loading, samples were boiled in Protein Loading Dye for 10 min at 95 °C. Proteins were separated by SDS-PAGE using 7.5% polyacrylamide gels and were transferred to a PVDF membrane, using a wet blotting system (Bio-Rad). Membranes were blocked by 5% non-fat milk in Tris buffered saline with Tween20 (TBST) (blocking buffer) for 2 h. Blots were incubated with primary antibodies [anti-FLAG (F1804, Sigma) at 1:1000 dilution; anti-β-actin (A1978, Sigma) at 1:4000 dilution in blocking buffer] overnight at 4 °C. The next day, after washing steps in TBST, the

membranes were incubated for 1 h with HRP-conjugated secondary anti-mouse antibody 1:20,000 (715-035-151, Jackson ImmunoResearch) in blocking buffer. The signal from detected proteins was visualized by ECL (Pierce ECL Western Blotting Substrate, Thermo Scientific) using a CCD camera (Bio-Rad ChemiDoc MP).

### In vitro transcription

sgRNAs were transcribed in vitro using TranscriptAid T7 High Yield Transcription Kit and PCR-generated double-stranded DNA templates carrying a T7 promoter sequence, following the manufacturer's protocol. PCR primers used for the preparation of the DNA templates are listed in Supplementary Data 1. sgRNAs were purified with the RNA Clean & Concentrator kit and reannealed (95 °C for 5 min, ramp to 25 °C at 0.3 °C/s). sgRNAs were quality checked using 10% denaturing polyacrylamide gels and ethidium bromide staining.

### Protein purification

WT SpCas9 was purified using pET-Cas9-NLS-6xHis (Addgene #62933)[39] plasmid, SpCas9-HF1 and SuperFi-SpCas9 were subcloned into that plasmid (details in Methods: Plasmid construction section and in Supplementary Information). The expression constructs of the SpCas9 variants were transformed into *E. coli* BL21 Rosetta 2 (DE3) cells, grown in Luria-Bertani (LB) medium at 37 °C for 16 h. 10 ml of this culture was inoculated into 1 l of growth media (12 g/l Tripton, 24 g/l Yeast, 10 g/l NaCl, 883 mg/l $NaH_2PO_4$ $H_2O$, 4.77 g/l $Na_2HPO_4$, pH 7.5) and cells were grown at 37 °C to a final cell density of 0.6 OD600, and then were cooled to 18 °C. The protein was expressed at 18 °C for 16 h following induction with 0.2 mM IPTG. Proteins were purified by a combination of chromatographic steps using the NGC Scout Medium-Pressure Chromatography Systems (Bio-Rad). Cells were centrifuged at $6000 \times g$ for 15 min at 4 °C. The cells were resuspended in 30 ml of Lysis Buffer (40 mM Tris pH 7.5, 500 mM NaCl, 20 mM imidazole, 0.5 mM TCEP) supplemented with Protease Inhibitor Cocktail (1 tablet/ 30 ml; complete, EDTA-free, Roche) and sonicated on ice. Lysate was cleared by centrifugation at $48,000 \times g$ for 40 min at 4 °C. Clarified lysate was bound to a 5 ml HisTrap™ High Performance Ni-Charged column (Cytiva). The resin was washed extensively with a solution of 40 mM Tris pH 7.5, 500 mM NaCl, 20 mM imidazole, and the bound proteins were eluted by a solution of 40 mM Tris pH 7.5, 250 mM imidazole, 150 mM NaCl. 10% glycerol was added to the eluted sample. The volume of the protein solution was diluted up to 60 ml with buffer (20 mM HEPES pH 7.5, 100 mM KCl, 1 mM DTT). Proteins were purified on a 5 ml HiTrap SP HP cation exchange column (GE Healthcare) and were eluted with 1 M KCl, 20 mM HEPES pH 7.5, 1 mM DTT. They were then further purified by size exclusion chromatography on a Superdex 200 10/300 GL column (GE Healthcare) in 20 mM HEPES pH 7.5, 200 mM KCl, 1 mM DTT and 10% glycerol. The eluted proteins were confirmed by SDS-PAGE and Coomassie brilliant blue R-250 staining, and they were stored at –20 °C.

### Determining the active SpCas9 quantity in solution

The quantification method was based on Liu et al.[42]. The quantity of active SpCas9 protein in solution was determined using EGFP target site 5, which has shown high cleavage activity with all three proteins tested, based on previous experiments. The measurement procedure is as follows: The target plasmid was incubated for an hour with protein-sgRNA complex, in different concentrations. Concentrations were determined by spectrophotometry (Nanodrop OneC), and then the target site containing the plasmid (6.2 nM) and the SpCas9 protein were mixed in a ratio between 1:0.8 and 1: 12, while the quantity of the sgRNA was twice that of the protein in each case. To terminate the cleavage reaction, EDTA solution (final concentration: 50 mM) was added to the reaction mix at 70 °C. Samples were ran on a 1% agarose gel. Following densitometry (GelQuantNET, BiochemLabSolutions.com), the ratio of intact plasmid and total DNA was calculated for

each sample. These values were plotted and fitted on a 'One-phase exponential decay function with time constant parameter' curve in Origin 2018. Based on the results of this experiment, the quantities of active SpCas9 variant in solution were calculated. The fact that SpCas9 has a one-fold turnover rate was also taken into consideration.

### Determining the cleavage rate of SpCas9 variants in vitro

First, two different solutions were made: (1) a target site containing plasmid solution and (2) an SpCas9-sgRNA master mix. Both solutions were diluted with the same cleavage buffer (final concentration: 20 mM HEPES pH 7.5, 200 mM KCl, 1 mM TCEP, 2% glycerol) and were pre-incubated at 37 °C before mixing. To trigger the cleavage reaction, the target plasmid solution was added to the SpCas9-sgRNA mixture. After mixing them the ratio of the target site containing plasmid and the active protein was 1:3. To terminate the cleavage reaction, EDTA (final concentration: 50 mM) was added to the reaction mix at 70 °C at different time points. First, we determined the approximate cleavage rate for every protein-target combination. Based on these preliminary results, we defined three cleavage rate categories for the time range in which sampling should take place: fast (3–30 s), medium (3–300 s), and slow (3–3600 s). To record the actual time of sampling precisely, a digital chronometer was attached to the pipette which can record time points in an application developed by us. Samples were then run on a 1% agarose gel. Following densitometry (GelQuantNET, Biochem-LabSolutions.com), the ratio of cleaved DNA was calculated for each sample. Experiments were performed in triplicates. These values were plotted and fitted on a 'One-phase exponential decay function with time constant parameter' curve in Origin 2018. All fitted curves are available in Supplementary Fig. 4, and the k values are available in Supplementary Data 5.

### Statistics

Differences between SpCas9 variants were tested by using RM one-way ANOVA and Dunnett's multiple comparisons test with a single pooled variance (Figs. 1b, g, 3a, b-on-target editing, 3c-MM1, Supplementary Fig 10b) or by using RM one-way ANOVA, with the Geisser-Greenhouse correction and (i) Dunnett's multiple comparisons test with individual variances computed for each comparison (Figs. 1a, c, 2e, f, 3b-bystander editing, 3c-MM2-3, 3d) or (ii) Tukey's multiple comparisons test with individual variances computed for each comparison (where the mean of each column was compared with the mean of every other columns: Supplementary Figs. 7c, 8b) in the cases where sphericity did not meet the assumptions of RM one-way ANOVA. Statistical tests were performed using GraphPad Prism 8 software. Test results are shown in Supplementary Data 6.

### Reporting summary

Further information on research design is available in the Nature Portfolio Reporting Summary linked to this article.

## Data availability

Expression vectors developed in this study are available from Addgene: pPIK16045_pX330_Flag-SuperFi-Cas9 (#184370), pAT15542_nCBE-SuperFi-Cas9(#184372), pAT15544_ nABE-SuperFi-Cas9 (#184374), pAT15543_dABE-SuperFi-Cas9 (#184373), pAT15546_nABE8e-SuperFi-Cas9 (#184376), pAT15545_dABE8e-SuperFi-Cas9 (#184375), pAT15547_PEmax-SuperFi-Cas9 (#184377). The deep sequencing data are available in NCBI Sequence Read Archive (accession number: PRJNA876837). Source data are provided with this paper.

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

## Acknowledgements

We thank György Váradi and the FACS core facility for their valuable help. We thank Ildikó Szűcsné Pulinka, Judit Szűcs, Vivien Karl, Ferencné Zájer, and Diána Szeregnyei for their excellent laboratory assistance. We thank Vanessza Laura Végi and Dóra Bokor for proofreading the manuscript and Zsuzsa Bartos and Zsófia Rakvács for their valuable help. The project was supported by grants K128188, K134968, and K142322 to E.W. and PD134858 to P.I.K. from the Hungarian Scientific Research Fund (OTKA) and grant 2018-1.1.1-MKI-2018-00081 to E.W. from the National Research, Development and Innovation Office. P.I.K. is a recipient of the János Bolyai Research Scholarship of the Hungarian Academy of Sciences (BO/764/20). S.L.K. was supported by grants VEKOP-2.1.7-17-2016-00383 and EFOP-3.6.3-VEKOP-16-2017-00009

from the Higher Education Institutional Excellence Program of Semmelweis University.

## Author contributions

P.I.K. and A.T. contributed equally to this work. S.L.K. analyzed NGS data, Z.L. purified the proteins and performed the in vitro experiments. P.I.K. and A.T. performed all other experiments, processed the data, and presented the results. P.I.K., A.T., and E.W. designed the experiments and interpreted the results. E.W. wrote the manuscript with input from the other authors.

## Funding

## Competing interests

The authors declare no competing interests.
