## [Peer Review File · Nature Communications]

Reviewers' Comments:

Reviewer #1:

Remarks to the Author:

In their manuscript, Kulcsar, Talas, and Welker study the on- and off-target activity of the recently described SuperFi-Cas9 variant in mammalian cell lines. In addition, they assess the on- and off-target activity of SuperFi-based base editor variants, and the on-target activity of a SuperFi-based prime editor variant. In all cases but one, they find that the SuperFi variant has greatly diminished on-target activity, while in that context retaining a favorable off: on-target activity ratio. In the one exception, in the context of the deaminase domain of ABE8e, on-target activity of the SuperFi variant is retained. While these are interesting data, this manuscript fundamentally represents a negative result and would not be of great interest to the gene editing community, as it neither positively identifies an improved fidelity variant nor provides mechanistic/prescriptive insight into why SuperFi does not have activity.

Comments:

1) The Bravo et al. paper is a very intriguing and comprehensive study of the structural basis of SpCas9 specificity. However, it is notable that the characterization of SuperFi-Cas9 in that study is purely in vitro. Therefore, it is not surprising that SuperFi-Cas9 was found in this manuscript to have low activity in cells. Bravo et al. tacitly acknowledge that SuperFi is a starting point, mentioning in the last sentence of their results that their data "constitute a proof of concept and provide a rationale for engineering improved variants of Cas9 using our structure." The manuscript from Kulcsar, Talas, and Welker would have been much stronger if the authors had built upon the model proposed by Bravo et al., and the authors' experience with engineering variants of Cas9, to build a SuperFi variant that does work in mammalian cells.

2) In the context of the authors' results, the title does not appear to tell the full story, as the "high fidelity" in mammalian cells comes at the price of on-target activity. The implication of the title is that SuperFi-Cas9 has high fidelity without losing on-target activity in mammalian cells.

2) Since the SuperFi variants do not appear to work well, the authors have an obligation to understand why there is low activity. Is it being expressed at the same level in mammalian cells? Have the transfections been optimized? Are there different kinetics?

3) In general, the use of reporter assays (EGFP disruption, BEAR, PEAR) is not sufficient for a study of off-target activity. It is reasonable for on-target screening for cleavage (as in Fig. 1a), but for base editor on-target activity (because of product purity and the potential for multiple edits to the same locus) and for prime editor on-target activity (due to precise and imprecise editing), amplicon sequencing is the gold standard. Off-target activity for all editors needs to be assessed by amplicon sequencing in multiple replicates. In addition, the use of BEAR and PEAR assays is not mainstream, so a short summary in the text or a figure panel explaining the assays would be useful.

4) Given that the on-target activity of the Cas9 variants tested in this study can be low, the plots of off: on target activity are oversimplifying the data. A different way of plotting, such as scatter plots (with dot colors corresponding to the different variants) may be more informative to visualize the tradeoff between on-target activity and off-target specificity.

Reviewer #2:

Remarks to the Author:

Kulcsár et al. characterize on-target and off-target activity of the recently developed SuperFi-Cas9 in mammalian cells for gene disruption, base editing, and prime editing. They show that while SuperFi-Cas9 has extremely high fidelity, it exhibits reduced on-target activity compared to WT and most first-generation IFNs. Overall, the manuscript provides a lot of useful data to the genome editing community, such as on- and off-target editing profiles, the compatibility of IFNs with truncated/extended gRNAs, BEs, and PEs. This study could benefit from deeper discussion of the data generated and some additional analysis.

- Line 69-70: It would be nice to discuss actual values here. For example, comparing means and distribution of SuperFi GFP disruption to WT and other Cas variants.
- Line 113-119: On average, how much did SuperFi reduce the base editing activity or bystander editing compared to other variants? Mentioning specific fold-changes where appropriate by give a clearer understanding of the relative level of editing events.
- Line 119 (Fig. 2b / Supplementary 6a): More analysis and discussion of the effects of SuperFi-Cas9 on ABE8e's base editing window would be useful here. Does the window shift or get narrower with SuperFi-Cas9? By how many nucleotides? For example, base editing at the genomic site Lig1 shows the most-edited base (A10) is outside of the canonical editing window and this may skew the A4-A8 editing on-target vs. bystander editing. SCN5a-intr also has an interesting editing profile where SuperFi and HeF both reduce PAM-proximal editing seen in the parental ABE8e.
- Are there any sequence determinants for the reduction in on-target activity for SuperFi-Cas9? Such as spacer GC content or internal gRNA stability?
- It is not clear in the main text which CBE variant was used (i.e. which deaminase domain). Since ABE8 is much better with SuperFi-Cas9 than ABE7, it may be the case that faster deaminases are required to be compatible with it. For example, a comparison between wild type rAPOBEC1 and evoAPOBEC.
- Do the authors have an explanation why ABE8 worked better than ABE7 with SuperFi? Or why prime editing failed?
- I did not see CBE off-target data, only for ABE.
- In certain figures such as 1a where SuperFi is being directly compared to wild-type, it may be helpful to show the data in another way (maybe in a supplemental figure) where the data point for the same sites are connected (to allow for direct visualization of the comparison between wild type and SuperFi). I would like to see if the sites with lower wild-type on-target activity have nearly zero activity with SuperFi or if it is random.
- I would like to see more discussion or data on how the variable activity between sites could be related to the explanation posited in the discussion, where they exclude the possibility of interference with cellular factors. Variable activity could suggest a non-universal cellular DNA damage response, especially regarding sequence context and histone accessibility.

Point-by point reply to the Reviewers' comments

Reviewer #1 (Remarks to the Author):

In their manuscript, Kulcsar, Talas, and Welker study the on- and off-target activity of the recently described SuperFi-Cas9 variant in mammalian cell lines. In addition, they assess the on- and off-target activity of SuperFi-based base editor variants, and the on-target activity of a SuperFi-based prime editor variant. In all cases but one, they find that the SuperFi variant has greatly diminished on-target activity, while in that context retaining a favorable off:on-target activity ratio. In the one exception, in the context of the deaminase domain of ABE8e, on-target activity of the SuperFi variant is retained. While these are interesting data, this manuscript fundamentally represents a negative result and would not be of great interest to the gene editing community, as it neither positively identifies an improved fidelity variant nor provides mechanistic/prescriptive insight into why SuperFi does not have activity.

Comments:

1) The Bravo et al. paper is a very intriguing and comprehensive study of the structural basis of SpCas9 specificity. However, it is notable that the characterization of SuperFi-Cas9 in that study is purely in vitro. Therefore, it is not surprising that SuperFi-Cas9 was found in this manuscript to have low activity in cells. Bravo et al. tacitly acknowledge that SuperFi is a starting point, mentioning in the last sentence of their results that their data constitute a proof of concept and provide a rationale for engineering improved variants of Cas9 using our structure. The manuscript from Kulcsar, Talas, and Welker would have been much stronger if the authors had built upon the model proposed by Bravo et al., and the authors experience with engineering variants of Cas9, to build a SuperFi variant that does work in mammalian cells.

While we agree with this point of the Reviewer, the generation, testing and appropriate characterization of new variants is out of the scope of the present study. We consider it should be a new project.

Although increased-fidelity variants have been prepared based on different rationales and/or approaches in different studies and they have different features, they are all similar to some extent. For example, they similarly trade activity for fidelity. Our experiments do not suggest that SuperFi or its mutations would be different from the “first generation” IFNs in this respect. We have recently generated and/or characterized over 30 IFNs with incrementally increasing activity/decreasing fidelity, and do not see the immediate demand for one more such nuclease. Thus, we did not aim to prepare a new SuperFi IFN with higher activity, which can actually be easily generated based on our characterization of the individual mutations of SuperFi regarding their contributions to fidelity/activity as we showed it in Fig. 2b,c in the revised manuscript.

2) In the context of the authors' results, the title does not appear to tell the full story, as the "high fidelity" in mammalian cells comes at the price of on-target activity. The implication of the title is that SuperFi-Cas9 has high fidelity without losing on-target activity in mammalian cells.

We have corrected the title to the following:

“SuperFi-Cas9 exhibits remarkable fidelity but severely reduced activity, yet works effectively with Adenine Base Editor 8e“

2) Since the SuperFi variants do not appear to work well, the authors have an obligation to understand why there is low activity. Is it being expressed at the same level in mammalian cells? Have the transfections been optimized? Are there different kinetics?

As it is shown in Supplementary Figure 1d and e in the revised MS, SuperFi seems to be expressed at the same level as the wild type Cas9. Furthermore, we routinely measure transfection efficiency by flow cytometry for all of our samples including amplicon sequencing to make sure the results are comparable. We have also successfully optimized the transfection efficiency in our earlier projects. We present figure 1d, e here as well:

Supplementary Figure 1 d,e:

d, Immunoblot analysis of the expression levels of SpCas9 nuclease variants in cell lysates of reporter N2a.dd-EGFP cells transfected with the indicated nuclease constructs. β -actin was used as a loading control for total protein amounts analyzed. *e*, Titration of expression plasmid amounts of wild-type SpCas9 and different IFN variants. Means are shown in case of each data point, error bars represent the standard deviation (SD) for triplicates; mean level of background EGFP loss in negative controls is represented by the red dashed line.

Our new experiments in the revised manuscript that show that SuperFi has low activity *in vitro* as well (Fig. 2a and Supplementary Fig. 5), provide an explanation for the low activity of SuperFi in cells. The fact that mutations of the RuvC loop are known to lead to decreased on-target activity (see ref 12 in the MS) may explain its generally decreased on-target activity. We present figure Fig. 2a here as well:

Figure 2a:

a, In vitro cleavage activities of the variants employing 8 targets of Figure 1a. Data is related to Supplementary Figure 5. The median and interquartile range are shown; data points represent the mean of the fitted k value triplicates. Summary of target and primer sequences and in vitro data are reported in Supplementary Table 1 and 5.

These new findings are shown in Figure 1d,e, 2a and Supplementary Figure 1d, e, and 5 and discussed on page 6 lines 123-128 and on page 11 lines 246-250.

3) In general, the use of reporter assays (EGFP disruption, BEAR, PEAR) is not sufficient for a study of off-target activity. It is reasonable for on-target screening for cleavage (as in Fig. 1a), but for base editor on-target activity (because of product purity and the potential for multiple edits to the same locus) and for prime editor on-target activity (due to precise and imprecise editing), amplicon sequencing is the gold standard. Off-target activity for all editors needs to be assessed by amplicon sequencing in multiple replicates. In addition, the use of BEAR and PEAR assays is not mainstream, so a short summary in the text or a figure panel explaining the assays would be useful.

We have inserted a short description of the two methods on page 7 lines 155-159. It reads as:

“For these experiments we utilized two recently-developed plasmid-based fluorescent assays, BEAR and PEAR which are suitable for assessing base and prime editing activity, respectively. Both assays are based on a reporter GFP sequence interrupted by an intron with an inactive splice donor site. When the splice site is corrected by the specific action of base editing or prime editing (indels do not correct the splice site) transiently fluorescent cells can be detected.”

While we agree with the Reviewer’s point, in fact, here we tried to assess whether the base editors have any activity and characterized only the ABE8e variants further by amplicon sequencing with multiple replicates, as only SuperFi-ABE8e showed any interpretable activity.

Regarding prime editing, the absence of editing activity by SuperFi-PE (PEmax 3) was also demonstrated on 13 endogenous targets as shown in Figure 4a (earlier in 2e).

In the revised manuscript we demonstrate the activity of SuperFi-ABE8e on more targets as assessed by amplicon sequencing (Fig. 3b [earlier Fig. 2b], Supplementary Fig. 7b). As per the Reviewer requested, we have demonstrated that SuperFi-ABE7 and SuperFi-CBE has diminished activity on genomic targets (in median 1% and 8%, respectively (Supplementary Fig. 7c, 8b), just as we had observed it earlier in the BEAR system. It is appropriate to assess off-target effect only in relation to substantial on-target activity. We did not find the activity of SuperFi-ABE7 and -CBE sufficiently high to call for an off-target characterization.

For a more convenient overview we present Supp. Fig. 7 and 8 here as well:

Supplementary Figure 7

a

c

b

Supplementary Figure 7. On-target activities of SuperFi base editor variants in the BEAR assay and on endogenous target sites

a, On-target activity of WT and base editor variants as indicated in the panel. The heatmap shows mean on-target base editing activity (as GFP positive cells) of triplicates on 20 different target sites. b, Editing efficiencies of ABE7, WT-ABE8e, and two IFN ABE8e base editor variants are shown on 16 genomic loci for assessing on-target base editing as measured by NGS. The diagrams show A to G (and in two cases C to G) conversion ratios at each adenine position for each target. As negative controls, base conversion for untransfected cells was measured (black with white dashes). Means are shown, error bars represent the standard deviation (SD) for triplicates (overlaid as white circles). c, Base editing activity of ABE7 variants on endogenous on-target sites, measured by NGS. The editing efficiency of adenines inside (A4-A8) and outside the editing window (bystander editing) is shown side by side, separated with a dashed line. Results are presented on a scatter dot plot, the median and interquartile ranges are shown; data points are plotted as open circles representing the means of triplicates. Statistical significance was assessed by RM one-way ANOVA, statistical details and p-values are available in Materials and Methods and in Supplementary Table 6.a-c, Target sequences, BEAR assay and NGS data are reported in Supplementary Tables 1–3.

Supplementary Figure 8 also includes data for SuperFi-evoCBE not covered here:

Supplementary Figure 8

Supplementary Figure 8. On-target activities of WT and SuperFi CBE base editor variants in the BEAR assay and on endogenous target sites

a-b, Different WT SpCas9 and SuperFi CBE base editor variants were tested: CBE3 and CBE4max40 (indicated as BE3 and BE4 in the figure, respectively) and evoCBE34. a, On-target activity of WT and base editor variants as indicated in the panel. The heatmap shows mean on-target base editing activity (as GFP positive cells) of triplicates on 14 different target sites. b, Base editing activity of CBE variants on endogenous on-target sites as measured by NGS. The editing efficiency of cytosines inside (C3-C7) and outside the editing window (bystander editing) is shown side by side, separated with a dashed line. Results are presented on a scatter dot plot, the median and interquartile ranges are shown; data points are plotted as open circles representing the means of triplicates. Statistical significance was assessed by RM one-way ANOVA, statistical details and p-values are available in Materials and Methods and in Supplementary Table 6. a-b, Target sequences, BEAR assay and NGS data are reported in Supplementary Tables 1–3.

4) Given that the on-target activity of the Cas9 variants tested in this study can be low, the plots of off: on target activity are oversimplifying the data. A different way of plotting, such as scatter plots (with dot colors

corresponding to the different variants) may be more informative to visualize the tradeoff between on-target activity and off-target specificity.

We agree with the Reviewer's concern, the off-target/on-target ratio is informative only when the on-target effect is substantial. We presented plots of off-target/on-target activity in Figure 1c, where we examined the mismatch tolerance of a variant only on targets on which it reached 70% of the WT activity (Fig. 1c and Supplementary Fig. 2a – the figures are copied below as well). The on-target values, along with the corresponding off-targets, are shown as a heatmap in Supplementary Figure 2, ensuring that the information the Reviewer wanted to see can readily be noticed.

However, we need to note, that the data in these figures may be not the best for visualizing the trade-off between on-target activity and off-target specificity. As we all know, IFNs exhibit decreased activity in a target-dependent manner. They show WT-like activity on several targets and no or strongly reduced activity on others. Here we test only those targets on which they have at least 70% of the WT activity, thus these figures do not give a clear picture of the trade-off. We demonstrate on-target activity on many more targets in Figure 1a and b (and Supplementary Figure 1b, c), and compare these data to IFNs with known activity/specificity. These data together give the picture the Reviewer wanted to see.

Figure 1c:

Figure 1c: Off-target activity of various SpCas9 variants as measured in EGFP disruption assay presented on a scatter dot plot. Only those data points are presented here for which the on-target activity exceeded 70%. Data points are from Supplementary Figure 2a. The median and interquartile ranges are shown; data points are plotted as open circles representing the mean of triplicates. Statistical significance was assessed by RM one-way ANOVA, statistical details and p-values are available in Materials and Methods and in Supplementary Table 6 (**** $p < 0.0001$).

Supplementary Figure 2a: Mismatch screen of nuclease variants with either perfectly matching sgRNAs (red to blue) or single mismatching sgRNAs (yellow to brown) presented on a heatmap. Grey boxes: not determined due to low on-target activity (< 0.2 normalized to WT). Target sequences, EGFP disruption, NGS data and statistical details are reported in Supplementary Tables 1–3 and 6.

		SpCas9 variant			
		WT	HF1	Hypa	SuperFi
Spacer sequences					
43	GAAGGGCATCGACTTCAAGG	1.00	1.00	1.00	1.00
	GAGGGCATCGACTTCAAGG	0.94	1.02	1.05	0.99
	GACGGGCATCGACTTCAAGG	1.12	1.03	1.02	0.99
	GATGGGCATCGACTTCAAGG	1.08	0.94	0.88	0.91
	GAGGGCATCGACTTCAAGG	1.14	0.93	0.80	0.91
	GAGGTGCATCGACTTCAAGG	1.03	0.92	0.82	0.91
	GAGGACATCGACTTCAAGG	0.92	0.89	0.88	0.92
	GAGGGGCATCGACTTCAAGG	0.97	0.92	0.87	0.92
	GAGGGTATCGACTTCAAGG	1.01	0.94	0.88	0.90
	GAGGGAAATCGACTTCAAGG	0.93	0.73	0.96	0.92
4	GGAGCGCACCATTCTTCTCA	1.00	1.00	0.91	0.92
	GGGGCGCACCATTCTTCTCA	1.00	0.69	0.88	0.92
	GGCGCGCACCATTCTTCTCA	1.05	0.81	0.81	0.92
	GTGGCGCACCATTCTTCTCA	0.90	0.91	0.88	0.92
	GGAGCGCACCATTCTTCTCA	0.79	0.60	0.91	0.92
	GGAGTGCACCATTCTTCTCA	0.93	0.89	0.76	0.92
	GGAGAGCACCATTCTTCTCA	0.84	0.80	0.90	0.92
	GGAGCGCACCATTCTTCTCA	0.70	0.79	0.42	0.91
	GGAGCGTACCATTCTTCTCA	0.96	0.88	0.66	0.92
	GGAGCGAACCATCTTCTTCA	0.92	0.98	0.81	0.92
1	GGGCACGGGCGAGTTGCCGG	1.00	1.00	0.91	0.92
	GTGCACGGGCGAGTTGCCGG	1.06	0.24	0.95	0.91
	GAGCACGGGCGAGTTGCCGG	0.92	0.78	0.95	0.90
	GGGACGGGCGAGTTGCCGG	1.12	0.70	0.83	0.93
	GGGTACGGGCGAGTTGCCGG	1.07	0.42	0.92	0.29
	GGGACGGGCGAGTTGCCGG	1.09	0.42	0.91	0.92
	GGGCACGGGCGAGTTGCCGG	0.99	0.61	0.79	0.92
	GGGCATGGGCGAGTTGCCGG	0.91	1.00	0.91	0.92
	GGGCACAAGGCGAGTTGCCGG	1.07	0.97	0.92	0.92
	GGGCACGGGCGAGTTGCCGG	1.00	0.98	0.91	0.92
5	GTCCCCCTCGAACTTCACTT	1.00	1.00	1.00	1.00
	GTCCCCCTCGAACTTCACTT	0.98	0.78	0.96	0.93
	GTCCCCCTCGAACTTCACTT	0.93	0.99	1.03	0.96
	GTCCCCCTCGAACTTCACTT	0.97	0.91	0.96	0.92
	GTCCCCCTCGAACTTCACTT	1.10	0.95	0.98	0.92
	GTCTCCCTCGAACTTCACTT	1.15	0.89	0.91	0.92
	GTCACTCCGAACTTCACTT	1.12	0.27	0.96	0.92
	GTCCCGCTCGAACTTCACTT	0.98	0.14	0.96	0.92
	GTCCCTCTCGAACTTCACTT	1.02	0.97	0.95	0.78
	GTCCCACTCGAACTTCACTT	0.96	0.98	0.91	0.92
9	GAAGTTTGAAGGCGACACCC	1.00	1.00	1.00	1.00
	GAGTTTGAAGGCGACACCC	0.95	0.12	0.98	0.92
	GACTTTGAAGGCGACACCC	0.90	0.13	0.65	0.92
	GATTTTGAAGGCGACACCC	0.87	0.57	0.67	0.92
	GAAAGTTGAAGGCGACACCC	0.91	0.90	0.91	0.92
	GAACTTGAAGGCGACACCC	0.86	0.98	0.95	0.92
	GAAATGAAGGCGACACCC	0.96	0.90	0.92	0.92
	GAAATTTGAAGGCGACACCC	0.78	0.21	0.21	0.92
	GAAATTTGAAGGCGACACCC	0.82	0.14	0.11	0.92
	GAAATTTGAAGGCGACACCC	0.88	0.97	0.95	0.92
3	GGTGGTGCAGATGAACTTCA	1.00	1.00	0.99	0.71
	GGGGTGCAGATGAACTTCA	0.96	0.55	0.71	0.92
	GGCGGTGCAGATGAACTTCA	0.89	0.87	0.88	0.92
	GGAGGTGCAGATGAACTTCA	0.94	0.93	0.84	0.92
	GGTGTGCAGATGAACTTCA	0.90	0.89	0.85	0.92
	GGTGTGCAGATGAACTTCA	0.89	0.71	0.63	0.92
	GGTGTGCAGATGAACTTCA	0.79	0.92	0.84	0.92
	GGTGTGCAGATGAACTTCA	0.55	0.97	0.92	0.92
	GGTATTCAGATGAACTTCA	0.62	0.32	0.28	0.92
	GGTATACAGATGAACTTCA	0.90	0.66	0.90	0.92
2	GAGCTGGACGGCGACGTAAA	1.00	1.00	0.91	0.20
	GGCTGGACGGCGACGTAAA	1.08	0.19	1.04	
	GGCTGGACGGCGACGTAAA	0.92	0.24	0.80	
	GTCTGGACGGCGACGTAAA	1.06	0.12	0.49	
	GAGTGGACGGCGACGTAAA	1.02	0.97	0.96	
	GAGTGGACGGCGACGTAAA	1.09	0.12	0.14	
	GAGTGGACGGCGACGTAAA	1.06	0.20	0.15	
	GAGCTGGACGGCGACGTAAA	0.99	0.95	0.92	
	GAGCTGGACGGCGACGTAAA	0.90	0.91	0.91	
	GAGCTGGACGGCGACGTAAA	0.95	0.66	0.67	

We hope that the presentations of the data in the revised manuscript satisfy the Reviewer's expectations.

We thank for the comments of Reviewer #1 which helped to improve our manuscript.

Reviewer #2 (Remarks to the Author):

Kulcsár et al. characterize on-target and off-target activity of the recently developed SuperFi-Cas9 in mammalian cells for gene disruption, base editing, and prime editing. They show that while SuperFi-Cas9 has extremely high fidelity, it exhibits reduced on-target activity compared to WT and most first-generation IFNs. Overall, the manuscript provides a lot of useful data to the genome editing community, such as on- and off-target editing profiles, the compatibility of IFNs with truncated/extended gRNAs, BEs, and PEs. This study could benefit from deeper discussion of the data generated and some additional analysis.

Line 69-70: It would be nice to discuss actual values here. For example, comparing means and distribution of SuperFi GFP disruption to WT and other Cas variants.

We have inserted the actual values on page 3 lines 80-85 of the revised manuscript. Now it reads as:

“SuperFi showed significantly lower average on-target activity than WT, Hypa- or SpCas9-HF1, although, on very few targets its activity approached that of the WT nuclease. Hypa- and SpCas9-HF1 reached about 88% and 83%, respectively, while SuperFi reached 15% of the WT activity in the disruption assay. (Percentages in brackets refer to median values throughout the article). Interestingly, on endogenous targets SuperFi exhibited lower normalized modifications (in median 4% in panel b). The difference likely reflects a more saturated condition in the disruption assay.”

Line 113-119: On average, how much did SuperFi reduce the base editing activity or bystander editing compared to other variants? Mentioning specific fold-changes where appropriate by give a clearer understanding of the relative level of editing events.

We have inserted the actual values on page 7-8, lines 161-167 of the revised manuscript. Now it reads as:

“When tested, SuperFi cytosine base editor 3 (CBE for short) and SuperFi adenine base editor 7.10 (ABE7 for short) exhibited strongly reduced base editing activity both in the BEAR assay (17% and 15%) and on endogenous targets (29% and 9%) normalised to WT CBE and WT ABE7, respectively) (Fig. 3a, Supplementary Fig. 7 and 8a-b). However, SuperFi-ABE8e exhibited more substantial activity both in the BEAR fluorescent assay (Fig. 3a and Supplementary Fig. 7a), and on genomic targets (Fig. 3b and Supplementary Fig. 7b), with 84% and 56% median editing normalised to the WT ABE8e, respectively.”

Line 119 (Fig. 2b / Supplementary 6a): More analysis and discussion of the effects of SuperFi-Cas9 on ABE8's base editing window would be useful here. Does the window shift or get narrower with SuperFi-Cas9? By how many nucleotides?

For example, base editing at the genomic site Lig1 shows the most-edited base (A10) is outside of the canonical editing window and this may skew the A4-A8 editing on-target vs. bystander editing. SCN5a-intr also has an interesting editing profile where SuperFi and HeF both reduce PAM-proximal editing seen in the parental ABE8e.

The Reviewer is right about these observations. On Lig1 SuperFi-ABE8e shows higher relative editing at a bystander position (A10) than ABE8e, while in contrast, with SCN5a-intr it shows lower relative editing at the same A10 position. The actual editing efficiency results from the interplay of a couple of factors, as discussed in ref. 1 in the manuscript. Thus, the conclusion on the editing window can be drawn after

synthetizing appropriate amount of data. Following the Reviewer's suggestion, we have generated more data (Fig. 3b [earlier Fig. 2b], Supplementary Fig. 7b). Our new experiments show very similar editing profile on 13 targets for ABE7, ABE8e and IFN-ABE8es (Supplementary Fig. 10a – the figure presented here as well).

Supplementary Figure 10a:

a, The editing window of the SuperFi-ABE8e base editor is not shifted in comparison to ABE7. Data analysed here are from Supplementary Figure 7b, extended with dead ABE8e and dead SuperFi-ABE8e data. (FANCFs1, EMX1 and HEK4 data were excluded because no dead ABE8e and dead SuperFi-ABE8e data were available for those target sites.) Each point represents the average for that base position.

This suggests that the editing window is not shifted. Rather, ABE8e shows substantial activity on positions that are outside of the ABE7's editing window. Usually this is not advantageous, since these are not the intended modifications. Here we refer to these positions as bystander. The effect of SuperFi and HeF on these bystander modifications likely results from the fact that by decreasing the activity of ABE8e, their effect is more apparent on the bystander modifications, as the positions in the editing window are more likely to have saturated effects.

Are there any sequence determinants for the reduction in on-target activity for SuperFi-Cas9? Such as spacer GC content or internal gRNA stability?

To reliably discern such features, the activity of SuperFi should be examined on thousands of target sequences which is out of the scope of the current study. Based on the relatively few target sequences examined here we could not discern any such effects.

It is not clear in the main text which CBE variant was used (i.e. which deaminase domain). Since ABE8 is much better with SuperFi-Cas9 than ABE7, it may be the case that faster deaminases are required to be compatible with it. For example, a comparison between wild type rAPOBEC1 and evoAPOBEC.

We fully agree with the Reviewer, and also think that it is a good idea. Indeed, our recent paper (ref. 32 in the MS) suggests exactly the same rationale as the Reviewer has suggested here, and it was the rationale behind testing ABE8e with SuperFi as well. We have inserted a short description of this rationale on page 8 lines 168-176. It reads as:

"In a former study we examined IFN-CBEs and IFN-ABEs on the same target sets using the BEAR assay. This experimental design facilitates the comparison of CBE and ABE base editing activities under conditions when SpCas9 encounters the same interactions with the targets, i.e.: the same extent/transiency of R-loop formation. These experiments revealed that higher fidelity IFNs such as evo- and HeFSpCas9 were not active in combination with ABE7 on targets on which they were active in combination with CBE, while they turned to be active with ABE8e which has an extremely fast deaminase variant. Altogether, these

findings suggested that the relation between the rate of deamination and the extent/transiency of R-loop formation determines how an IFN alters on- and off-target activity of base editors (the factors affecting base editors' activity is summarized in ref 34). Along this line of thought, we tested whether using a faster rAPOBEC deaminase variant, such as the one present in evoCBE would increase the base editing activity of SuperFi."

We have executed the suggested experiments and found that SuperFi-evoCBEmax shows slightly higher activity than SuperFi-CBEmax (Supplementary Fig. 8 – the figure is presented below as well). However, to have a really useful tool, an even faster deaminase seems to be necessary. The data are shown in Supplementary Figure 8 (shown below as well) and discussed on page 8 lines 178-183.

Thanks for the idea of testing evoAPOBEC.

Supplementary Figure 8:

Supplementary Figure 8. On-target activities of WT and SuperFi CBE base editor variants in the BEAR assay and on endogenous target sites

a-b, Different WT SpCas9 and SuperFi CBE base editor variants were tested: CBE3 and CBE4max40 (indicated as BE3 and BE4 in the figure, respectively) and evoCBE34. a, On-target activity of WT and base editor variants as indicated in the panel. The heatmap shows mean on-target base editing activity (as GFP positive cells) of triplicates on 14 different target sites. b, Base editing activity of CBE variants on endogenous on-target sites as measured by NGS. The editing efficiency of cytosines inside (C3-C7) and outside the editing window (bystander editing) is shown side by side, separated with a dashed line. Results are presented on a scatter dot plot, the median and interquartile ranges are shown; data points are plotted as open circles representing the means of triplicates. Statistical significance was assessed by RM one-way ANOVA, statistical details and p-values are available in Materials and Methods and in Supplementary Table 6. a-b, Target sequences, BEAR assay and NGS data are reported in Supplementary Tables 1–3.

Do the authors have an explanation why ABE8 worked better than ABE7 with SuperFi?

We have tested SuperFi based on the rationale that the Reviewer also suggested in the previous comments. The deaminase variant of ABE8e is much faster than ABE7. We have made it clearer in the revised MS on page 7-8 lines 159-176.

Or why prime editing failed?

Actually, it is not 100% clear. It failed even on targets where SuperFi works as a nuclease. We think it is related to the failure of SuperFi to work as a H840A nickase on targets where it works as a nuclease. We performed an EGFP disruption experiment with D10A and H840A SuperFi-, HeF- and WT-Cas9 paired nickases and found that these IFNs have little to no activity with the H840A mutation (that is present in prime editors).

We have also tested in a double nickase disruption assay if the inactivity is related to the combined effect of the seven SuperFi and the R221K, N394K mutations present in PEmax. These lysine mutations are supposed to increase the nuclease activity, and in PEmax these mutations may increase the nickase activity of SpCas9. Interestingly, the presence or absence of the R221K, N394K mutations did not affect the nickase activity of WT prime editors here. Moreover, neither SuperFi-PE, nor SuperFi-PEmax showed nickase activity on any of the target pairs (Supplementary Fig. 10f). Thus, our data do not support that the inactivity of SuperFi-prime editors is due to the presence of the R221K, N394K mutations.

These data are shown in Figure 4b and Supplementary Figure 10f (presented below here as well) and discussed on page 10, lines 215-231.

Figure 4b:

On-target disruption activities of nickase SpCas9 variants with paired sgRNAs. The heatmap shows the mean on-target modifications (indels) of three parallel transfections. (right) +/- indicate the activity of the nuclease variants with the corresponding sgRNAs. Summary of target and primer sequences, EGFP disruption and NGS data are reported in Supplementary Table 1-3.

Supplementary Figure 10f:

f

(left) On-target disruption activities of nickase SpCas9 variants with paired sgRNAs. The heatmap shows the mean on-target modifications (indels) of three parallel transfections. (right) +/- indicate the activities of the nuclease variants with the corresponding sgRNAs. Target sequences and EGFP disruption data are reported in Supplementary Tables 1–3.

I did not see CBE off-target data, only for ABE.

We have characterized the IFN-ABE8e variants for their off-target activities (see Fig. 3c, d and Supplementary Fig. 9), since they have substantial on-target activities (in the BEAR assay in median 47%, on genomic targets in median 35%). By contrast, the activity of SuperFi-ABE7 and -CBE is very low, on genomic targets in median 1% and 8%, respectively. Therefore, we thought there was no point in assessing their off-target activity (Supplementary Fig. 7c, 8b).

In certain figures such as 1a where SuperFi is being directly compared to wild-type, it may be helpful to show the data in another way (maybe in a supplemental figure) where the data point for the same sites are connected (to allow for direct visualization of the comparison between wild type and SuperFi). I would like to see if the sites with lower wild-type on-target activity have nearly zero activity with SuperFi or if it is random.

Actually, this type of presentation of the data is available in Supplementary Figure 1b and c, where the corresponding targets are aligned under each other. The figures are copied here as well:

Supplementary Figure 1b-c:

b, On-target EGFP disruption data of WT SpCas9 and three IFN variants, as indicated in the panels, on 24 EGFP target sites. *c*, On-target activity of WT SpCas9 and three IFN variants, as indicated in the panels, across 26 endogenous sites as measured by NGS. *b-c*, Means are shown, error bars represent the standard deviation (SD) for triplicates (overlaid as white circles). (*b*) Level of background EGFP loss is indicated by a red dashed line (average of the percentage of dead SpCas9 controls from all target sites). (*c*) In case of endogenous target sites, the measured background level is available in Supplementary Table 3. Target sequences, EGFP disruption, NGS data and statistical details are reported in Supplementary Tables 1–3 and 6.

I would like to see more discussion or data on how the variable activity between sites could be related to the explanation posited in the discussion, where they exclude the possibility of interference with cellular factors. Variable activity could suggest a non-universal cellular DNA damage response, especially regarding sequence context and histone accessibility.

The variable activity between sites is definitely influenced by cellular factors in the case of WT SpCas9. What we have tried to argue is that the decreased activity of SuperFi relative to WT on various targets is less likely to be due to cellular factors. The *in vitro* data demonstrate a similarly low activity of SuperFi, which convincingly supports this view. These data are shown below and in Figure 2a (and Supplementary Fig. 5).

Figure 2a:

a
in vitro activity of SpCas9 variants

a, *In vitro* cleavage activities of the variants employing 8 targets of Figure 1a. Data is related to Supplementary Figure 5. The median and interquartile range are shown; data points represent the mean of the fitted *k* value triplicates. Summary of target and primer sequences and *in vitro* data are reported in Supplementary Table 1 and 5.

We thank the Reviewer for the comments and suggestions that helped to improve our manuscript.

Reviewers' Comments:

Reviewer #1:

Remarks to the Author:

In their revised manuscript, Kulcsar, Talas, and colleagues have presented data characterizing the properties of the recently described SuperFi-Cas9 nuclease. This revised manuscript is much clearer to me than the initial manuscript, and I now see its relevance to the gene editing field. There remain some comments, regarding both the tone of the discussion and the need to add to the NGS data.

Comments:

1) A general comment for all of the NGS data in the paper is that allele tables should be included to give the reader a sense of the DNA modifications that are being identified. This is especially true for the ABE data re: editing window. In addition to the summary figure on editing window (Supplementary Figure 10a), the authors should also provide allele tables showing the frequencies of the alleles that are most frequently edited, since those data in a way more illustrative of editing window than the pooled (unphased) data. These data are also useful to understand how the summary data were calculated, since there are many ways of filtering edited sequences from unedited sequences based on locations and nature of edits. The methods describe how indels were filtered but not how base edits were filtered.

2a) The EGFP disruption assay can show loss of GFP signal solely due to binding (i.e. CRISPRi), as opposed to truly creating indels. The authors should comment on this possibility on p.3 as a reason why indels observed by NGS (albeit on different sites) could be lower than those noted by EGFP disruption.

2b) Given the differences in EGFP disruption vs. NGS that the authors have noted, the analysis of the 21G guides in Figure 1gh and Supplementary Figure 4 needs to also be done via NGS on endogenous sites.

3) The on-target activity of SuperFi on the four sites in Supplementary Figure 3a is much higher than would be expected from the data shown from other gRNAs in Figure 1/Supplementary Figure 1c. This certainly makes the GUIDE-seq analysis relevant, since it would not be without on-target activity, but the authors should comment on whether they purposefully chose sites with relatively high SuperFi on-target activity.

4) The tone of the discussion is a bit harsh re: Bravo et al. and should be written with less judgment and more objectivity. For example, the first sentence of the discussion is overly editorial and should be rewritten. Bravo et al. is a high profile study, but it is not appropriate to state that it does not "live up to the high expectations." Similarly with the last line of p.11 (lines 251-252) and lines 257-258, which has a vaguely accusatory tone. It would be more appropriate to point out that the study of a larger set of gRNAs/target sites has shown that both the in vitro and in cell activity of SuperFi is lower than its counterparts.

Minor Comments:

- Reference 2 (which has a poorly formatted citation in the reference list on p. 38) is more of an editorial than a review and should not be cited here.
- Figure 2 does not seem to be significant enough for the main text (at least on its own and especially Figure 2a), but it is important to the manuscript. It fits better as a supplemental figure. Furthermore, the results sections describing the data in Figure 2 are rather sparse and probably should be included in the first results subsection "SuperFi shows strongly reduced by high fidelity gene editing." I am assuming the authors have included it in the main text since they have reached ten supplementary figures, but perhaps some figure shuffling could be done.

Reviewer #2:

Remarks to the Author:

The authors have done an excellent job addressing my concerns, and the additional experiments really elevate the manuscript in my opinion. I think the manuscript is suitable for publication in Nature communications.

Reviewer #1 (Remarks to the Author):

In their revised manuscript, Kulcsar, Talas, and colleagues have presented data characterizing the properties of the recently described SuperFi-Cas9 nuclease. This revised manuscript is much clearer to me than the initial manuscript, and I now see its relevance to the gene editing field. There remain some comments, regarding both the tone of the discussion and the need to add to the NGS data.

We appreciate these positive comments and thank the Reviewer for the supportive opinion.

Comments:

1) A general comment for all of the NGS data in the paper is that allele tables should be included to give the reader a sense of the DNA modifications that are being identified. This is especially true for the ABE data re: editing window. In addition to the summary figure on editing window (Supplementary Figure 10a), the authors should also provide allele tables showing the frequencies of the alleles that are most frequently edited, since those data in a way more illustrative of editing window than the pooled (unphased) data. These data are also useful to understand how the summary data were calculated, since there are many ways of filtering edited sequences from unedited sequences based on locations and nature of edits. The methods describe how indels were filtered but not how base edits were filtered.

We thank the Reviewer for raising this point. For all ABE variant editing window experiments, we have provided allele frequency tables to illustrate the editing windows of the variants from a different angle. The methodology for summary data calculations for base editing analysis has already been described in the Materials and Methods section under the subheading “Indel, base and prime editing analysis by next-generation sequencing (NGS)”, right after the description of how we filtered indel frequency. According to the Reviewer’s feedback, it was not fully comprehensible, thus we have revised this text to make it clearer. Also, we have provided a description under the same subheading on how allele frequency tables were generated. The allele frequency tables are presented in Supplementary Table 7, and referred to in the main text with a new sentence added in page 9 line 209: “...for allele frequency tables see Supplementary Table 7).”.

2a) The EGFP disruption assay can show loss of GFP signal solely due to binding (i.e. CRISPRi), as opposed to truly creating indels. The authors should comment on this possibility on p.3 as a reason why indels observed by NGS (albeit on different sites) could be lower than those noted by EGFP disruption.

2b) Given the differences in EGFP disruption vs. NGS that the authors have noted, the analysis of the 21G guides in Figure 1gh and Supplementary Figure 4 needs to also be done via NGS on endogenous sites.

According to the Reviewer’s request we have included a sentence on page 3 lines 86-88 to indicate that binding to the targets in itself results in little contribution to the loss of the GFP signal, as follows: “Inhibiting transcription by the binding of SpCas9 to its targets contributes little to this effect in these EGFP disruption experiments, as demonstrated by the dead (inactive) SpCas9 control.”

Regarding the analyses of the 21G-sgRNAs, the Reviewer’s logic is likely the following: since the EGFP disruption assay is a more saturated condition, it can be supposed that if we had used a less saturated condition, we may have actually revealed that SuperFi has severely decreased activity with 21G-sgRNAs, alike other IFNs.

We have performed the experiment as suggested by the Reviewer. Due to the extremely low activity of SuperFi, which is around 5% in Figure 1b, we would need to screen 20 randomly selected targets with 20G-sgRNAs to find one with an activity that could make it reasonable to examine whether 21G-sgRNAs diminish the activity of SuperFi or not. This seemed to be a disproportionate amount of work, so we rather relied on the results of our recent study to select target sequences (Kulcsár et al., BioRxiv

<https://doi.org/10.1101/2022.08.16.504097>). In that study we found that target sequences have different cleavability, and we chose those targets that can also be cut by IFNs of lower average activity, specifically by evoSpCas9 in the cited study. We selected 6 endogenous targets, which were expected to be cleaved by SuperFi and also by evoSpCas9 with 20G-sgRNAs. However, they were only cut by SuperFi with 21G-sgRNAs, albeit with lower efficiency on some targets than we saw in the EGFP disruption experiments, which is somewhat in line with the Reviewer's expectation.

The results of the experiments are depicted in Figure 2c and d and discussed on page 6 lines 140-144:

3) The on-target activity of SuperFi on the four sites in Supplementary Figure 3a is much higher than would be expected from the data shown from other gRNAs in Figure 1/Supplementary Figure 1c. This certainly makes the GUIDE-seq analysis relevant, since it would not be without on-target activity, but the authors should comment on whether they purposefully chose sites with relatively high SuperFi on-target activity.

We thank the Reviewer for pointing this out. We have clarified this issue on page 6 lines 101-104 as follows: "To have meaningful results for these assays we selected targets on which SuperFi (similar to these higher fidelity IFNs) is expected to exhibit a reasonable on-target activity. This rationale is based on our recent study²⁵ showing that targets have different cleavability, and low activity/high fidelity IFNs can cleave only high cleavability targets.

4) The tone of the discussion is a bit harsh re: Bravo et al. and should be written with less judgment and more objectivity. For example, the first sentence of the discussion is overly editorial and should be rewritten. Bravo et al. is a high profile study, but it is not appropriate to state that it does not "live up to the high expectations." Similarly with the last line of p.11 (lines 251-252) and lines 257-258, which has a vaguely accusatory tone. It would be more appropriate to point out that the study of a larger set of gRNAs/target sites has shown that both the in vitro and in cell activity of SuperFi is lower than its counterparts.

We fully agree with this point of the Reviewer, and it is exactly what we attempted to do. The genome editing community expects us to provide an explanation why SuperFi shows such a low activity, just as the Reviewer also warned us in the first-round review, in the second major comment: "Since the SuperFi variants do not appear to work well, the authors have an obligation to understand why there is low activity". We simply started to discuss this issue in the first sentence of the discussion. Actually, in this sentence we did not write that the Bravo et al. study does not live up to the high expectations, as the Reviewer seems to have misunderstood this text. We wrote that "SuperFi did not completely live up to the high expectations generated by the study of Bravo et al", which is a fully objective statement. SuperFi was exposed as a "next generation" increased-fidelity variant with uncompromised on-target activity. This claim has really generated high expectations, the article was downloaded more than 100,000 times in a few days and countless commentaries discussed its development with titles such as: "New SuperFi-Cas9 Increases Fidelity Without Sacrificing Speed", "SuperFi-Cas9 to the rescue: gene editing just got safer", "SuperFi-Cas9: High Fidelity Meets High Activity", etc. Even Jack Bravo described their finding with the claim "SuperFi-Cas9 is like a self-driving car that has been engineered to be extremely safe, but it can still go at full speed". We do not think that we need to gather more titles and comments to show the high expectations that the introduction of SuperFi has generated. Unfortunately, SuperFi can go with just about 10% of full speed, meaning that it does sacrifice speed. So, we think that the first sentence of the Discussion part of our manuscript is absolutely correct, and correctly exposes the main issue we discuss in the next paragraph.

Likewise, we respectfully disagree with the Reviewer that our sentence at lines 268-270 in the original manuscript, ("Thus, a likely explanation is that the single target used in ref. 23 may happen to be one of the few targets on which SuperFi can demonstrate an activity rate close to that of WT-SpCas9.") would have a

“vaguely accusatory tone”. We investigated SuperFi on more than 50 targets and showed that it worked with WT-like activity on a very few ones only (about 4–10% of the targets investigated). SuperFi did show WT-like activity on the single target that Bravo et al. examined. Thus, apparently, it belongs to the few targets on which SuperFi shows higher activity. It is what we stated here. It is rather an objective statement of a fact than accusation, and we can hardly provide any other explanations for this finding based on the available data including ours.

We also cannot agree with the Reviewer’s opinion on the sentence in lines 274-277: “Somehow, the rationale underlying its development did not seem to be manifested in the features of SuperFi, as it has been revealed in these experiments.” In our view it is also just an objective conclusion drawn from the experiments. Bravo et al. suggested that SuperFi does not trade on-target activity for fidelity. Furthermore, from their rationale it follows that SuperFi should not show increased fidelity with mismatches that are not located in positions 18–20. Thus, these are the specific features that SuperFi should possess, but in fact it does not seem to have these features. Clearly, SuperFi does not have "high fidelity" because the mutations disrupt the interaction between the seven amino acids and the mismatching nucleic acid residues seen in the 18-20 MM structure. These experiments rather point to an important general role of the RuvC loop in the on-target activation of SpCas9. Since our sentence apparently could be misunderstood, we have now complemented the statements with an explanatory sentence to indicate its objectivity on page 12 lines 275-277 as follows: “...the RuvC loop mutations in the engineered variant do not specifically act on the targets with 18–20 mismatches, leaving on-targets and targets with mismatches at other positions unaffected.”

Since we have paid specific attention to avoid any harsh or accusatory tone, we are a bit surprised by this latter comment of the Reviewer. Exactly for this reason, i.e. to avoid the impression of a harsh and accusatory tone, in the manuscript we did not discuss the following arguments that we think are objective facts:

(i) Characterizing the activity of an increased-fidelity SpCas9 variant only on a single target is completely inappropriate. Even for the highest fidelity variants that have less average activity than SuperFi we have found targets on which their activities approach that of the WT (Kulcsár et al. *Genome Biology*, 2017 and Kulcsár et al., *Nature Communications*, 2021). It is a general feature of increased-fidelity SpCas9 variants that while they work with WT-like activity on some targets, they show no activity on others. Thus, by investigating only a single target, one may learn basically nothing about the general activity of an IFN, thus it is definitely a misleading approach.

(ii) We also did not point out that Bravo et al. based the development of SuperFi on a completely erroneous rationale. They argued that the RuvC loop is not resolved in any SpCas9 structures, and took this information as a proof to support that it is not involved in any interactions during on-target activation, so by mutating the RuvC loop residues, the on-target activity will not be affected. However, parts of the RuvC loop that contain some of the mutated residues of SuperFi are resolved in an on-target structure (Nishimasu et al., *Cell*, 2014). Furthermore, even at the time when Bravo et al. executed their study, experimental data were available to indicate that mutating one of the SuperFi residues diminishes the overall on-target activity of the SpCas9 variants (Tyr1013, Kulcsár et al, *Nature Communications*, 2021.) It is unfortunate that these flaws did not become apparent in the Bravo et al. study.

We hope that we have convinced the Reviewer that we have made hard efforts to objectively demonstrate and discuss our results relevant to the characterization of SuperFi, while really trying to avoid any harsh or disrespectful manner towards a research fellow in this field. Especially because we agree with the Reviewer that Bravo et al. have published a very intriguing and comprehensive study on the structural basis of SpCas9 specificity, and only some of the interpretations of the structures and the SuperFi story seems to be inadequate.

Minor Comments:

- Reference 2 (which has a poorly formatted citation in the reference list on p. 38) is more of an editorial than a review and should not be cited here.

We have removed this citation.

- Figure 2 does not seem to be significant enough for the main text (at least on its own and especially Figure 2a), but it is important to the manuscript. It fits better as a supplemental figure. Furthermore, the results sections describing the data in Figure 2 are rather sparse and probably should be included in the first results subsection "SuperFi shows strongly reduced by high fidelity gene editing." I am assuming the authors have included it in the main text since they have reached ten supplementary figures, but perhaps some figure shuffling could be done.

We thank the Reviewer for this suggestion, and we basically agree with it. We have shuffled the panels of Figure 1 and 2 and the corresponding texts. Now Figure 1 contains all the on-target and off-target data including *in vitro* ones. Figure 2 contains data on SuperFi's further characterization in terms of the effects of the individual mutations and data on its 21G-sgRNA activity. The previous Fig. 1g and 1h are now included as Fig. 2a-b, and the new results on endogenous loci with 21G-sgRNAs are presented as Fig. 2c-d.

Finally, we thank the Reviewer for the time spent on our manuscript and for the valuable suggestions to help us improve our manuscript. We think that it has substantially improved due to the Reviewer's useful comments. We really appreciate it!

Reviewer #2 (Remarks to the Author):

The authors have done an excellent job addressing my concerns, and the additional experiments really elevate the manuscript in my opinion. I think the manuscript is suitable for publication in Nature communications.

We thank the Reviewer for the positive comment and for all the suggestions in the review process that helped us to improve our manuscript.

Reviewers' Comments:

Reviewer #1:

Remarks to the Author:

The authors have again improved the manuscript and have addressed all of my concerns but one. I still believe the first line in the discussion stating that the Bravo paper does "not completely live up to the high expectations generated" is too editorial for a manuscript (as opposed to a commentary, news article, or presentation). That said, if the editor is comfortable with the tone, I am sympathetic to the author's rebuttal.

In any case, this manuscript will be a useful contribution to the gene editing community and to the readership of Nature Communications.

Reviewer #1 (Remarks to the Author):

The authors have again improved the manuscript and have addressed all of my concerns but one. I still believe the first line in the discussion stating that the Bravo paper does "not completely live up to the high expectations generated" is too editorial for a manuscript (as opposed to a commentary, news article, or presentation). That said, if the editor is comfortable with the tone, I am sympathetic to the author's rebuttal.

In any case, this manuscript will be a useful contribution to the gene editing community and to the readership of Nature Communications.

We intended to write that the SuperFi variant is the one that "did not fully live up to the high expectations generated", not the Bravo paper as a whole. As the sentence was interpreted by both the reviewer and the editorial team as referring to the study, we have corrected the ambiguous sentence.

We thank the Reviewer for the positive comment and for all the suggestions in the review process that helped us to improve our manuscript.